# Extensive gut virome variation and its associations with host and environmental factors in a population-level cohort

Suguru Nishijima [1,2,15,16] ✉, Naoyoshi Nagata [3,4,16] ✉, Yuya Kiguchi[1,5], Yasushi Kojima[4], Tohru Miyoshi-Akiyama[6], Moto Kimura[7], Mitsuru Ohsugi[8,9], Kohjiro Ueki[10], Shinichi Oka[11], Masashi Mizokami[12], Takao Itoi[13], Takashi Kawai[3], Naomi Uemura[14] & Masahira Hattori[1,5]

Indigenous bacteriophage communities (virome) in the human gut have a huge impact on the structure and function of gut bacterial communities (bacteriome), but virome variation at a population scale is not fully investigated yet. Here, we analyse the gut dsDNA virome in the Japanese 4D cohort of 4198 deeply phenotyped individuals. By assembling metagenomic reads, we discover thousands of high-quality phage genomes including previously uncharacterised phage clades with different bacterial hosts than known major ones. The distribution of host bacteria is a strong determinant for the distribution of phages in the gut, and virome diversity is highly correlated with anti-viral defence mechanisms of the bacteriome, such as CRISPR-Cas and restriction-modification systems. We identify 97 various intrinsic/extrinsic factors that significantly affect the virome structure, including age, sex, lifestyle, and diet, most of which showed consistent associations with both phages and their predicted bacterial hosts. Among the metadata categories, disease and medication have the strongest effects on the virome structure. Overall, these results present a basis to understand the symbiotic communities of bacteria and their viruses in the human gut, which will facilitate the medical and industrial applications of indigenous viruses.

Large numbers of diverse bacterial viruses (phages) reside in the human gut, and the indigenous phage community (virome or phageome) greatly affects the structure and function of the bacterial community (bacteriome) by lysing bacterial cells and facilitating horizontal gene transfer[1–4]. The most dominant members of the gut virome are double-stranded DNA (dsDNA) phages, and their number is estimated to be comparable to that of bacteria[1].

Recent viral metagenomic studies have identified highly abundant and prevalent clades among human gut dsDNA phages[5–11], such as crAss-like phages[6,7] including crAssphage[5], Lak phage[8] and Gubaphage[9]/Flandersviridae[10]. The genome sequences of these phages provide insights into their functional potential, unique

biology and ecology in the human gut[7,8]. Other studies targeting the virome structure have revealed its high inter-individual diversity[12,13], differences between populations[14], and associations with several host factors[15,16]. Furthermore, altered gut virome structure has been associated with various diseases, such as inflammatory bowel disease, type 2 diabetes, and colorectal cancer[17–19], suggesting a possible role for the altered virome in these diseases. However, these studies were conducted in relatively small-scale cohorts ($n < 1000$), or used heterogeneous samples collected from independent studies using different methodologies, serious constraints limiting the understanding of the virome variation and its associations with various host and environmental factors. Moreover, the DNA

amplification step frequently used in the previous studies could preferentially amplify certain types of viruses (e.g. circular single-stranded DNA phages) and systematically bias the viral profile[1]. To date, there have been no population-level studies that have collected and processed faecal samples uniformly without DNA amplification in order to analyse gut virome variations.

In this study, we present a large-scale analysis of human gut viral profiles obtained from whole (bulk) gut metagenomes uniformly collected from 4198 deeply phenotyped individuals. A newly developed pipeline revealed thousands of high-quality dsDNA phage genomes, including highly abundant but uncharacterised phage clades in the gut. Further comparative analysis of the virome with the bacteriome showed high correlations of their diversities, close interactions between phages and predicted bacterial hosts, and significant associations between the defense mechanisms of the bacteriome and virome diversity. Finally, a comprehensive association analysis with various host and environmental factors uncovered a large number of intrinsic and extrinsic factors significantly associated with the structure of the virome.

## Results

### Construction of a phage genome catalogue in the Japanese 4D microbiome cohort

We collected faecal samples from 4198 Japanese individuals (66.4 ± 12.6 years old (mean ± s.d.) [143 young (<40 years old), 2102 middle-aged (40–70 years old), and 1953 elderly (≥70 years old)]; proportion of males, 59%) in the Japanese 4D (Disease, Drug, Diet, Daily life) microbiome project and performed whole shotgun metagenomic sequencing (Supplementary Data 1)[20]. Intrinsic and extrinsic factors (e.g. age, sex, body mass index [BMI], lifestyle, dietary habits, diseases, and medications) were exhaustively collected from these individuals through self-report questionnaires, face-to-face interviews, and medical records (Supplementary Data 2).

To establish a dsDNA phage genome catalogue using the whole metagenomes, we developed a pipeline in which candidate phage genomes were identified by detecting phage genome-specific signatures, such as the presence of virus hallmark genes and the absence of bacterial essential genes (**Methods**, Supplementary Fig. 1a, Supplementary Data 3). Since the majority of reads in whole metagenomes derives from non-phage entities, such as bacterial chromosomes and plasmids[21], we used relatively strict criteria and designed the pipeline to reduce the false positives as much as possible (**Methods**). Comparison between our pipeline and other virus-detection tools[22–28] showed that the true positive rate of our pipeline (81.5%) was comparable or slightly lower than other pipelines' (40.2–99.7%), whereas the false positive rate (0.4%) was considerably lower than others' (2.7–54.2%) (Supplementary Fig. 1b, c). These results suggest that our pipeline, with the highest specificity at the cost of slightly lower sensitivity, was likely to be appropriate to construct a high-quality phage catalogue with minimum contamination by non-phage sequences.

Applying the pipeline to the 4198 whole metagenomes, we identified 1125 complete and 3584 draft (>70% completeness) phage genomes[29]. Quality assessment by CheckV revealed that 2819 (59.9%), 1836 (39.0%) and 54 (1.1%) of the genomes were complete/high-quality, medium-quality and low-quality, respectively (Supplementary Fig. 1d, Supplementary Data 4). To assess to what extent this catalogue covered the dsDNA phages in the human gut, we sequenced virus-like particles (VLPs) in additional 24 faecal samples (**Methods**, Supplementary Data 5) and mapped the VLP reads to the catalogue, confirming that more than half of the VLP reads (57.1%), on average, were mapped to the catalogue. This value was comparable to that from another phage genome catalogue constructed using more sensitive tool, VIBRANT (58.5%), suggesting that the slightly lower sensitivity of our pipeline had little effect on the viral coverage of the catalogue (Supplementary Note). The unmapped VLP reads might derive from

low abundant phages in the gut or those difficult to be assembled with short reads (Supplementary Note).

Clustering of the 4709 phage sequences with >95% sequence similarity generated 1347 viral operational taxonomic units (vOTUs)[29] (corresponding to the species level) (Fig. 1a and Supplementary Fig. 2a, Supplementary Data 4). The largest vOTU (vOTU_974) was composed of 461 genomes and represented a cluster of crAssphage, the most prevalent and abundant phage in the human gut[5]. Comparative analysis of the vOTUs with known phage genomes in RefSeq and several databases recently published[9,15,30,31] revealed that only 0.67–44.6% of the sequences we found were aligned with known sequences (>95% identity) (Supplementary Fig. 3a–e). 667 of the vOTUs were not aligned to any known genomes in the databases, suggesting that they are novel genomes first identified in this study (Supplementary Fig. 3f). The majority of the phages in the catalogue were predicted to belong to the order *Caudovirales* (n = 598), among which *Siphoviridae* (n = 291), *Podoviridae* (n = 213), and *Myoviridae* (n = 94) were most abundant (Fig. 1b).

Host prediction of the vOTUs using CRISPR spacers (**Methods**) revealed that the most common phylum of the predicted hosts was Firmicutes (n = 413), followed by Bacteroidetes (n = 271) and Actinobacteria (n = 118) (Fig. 1c). At the genus level, the commonly assigned hosts were *Bacteroides* (n = 176), *Ruminococcus* (n = 128), *Blautia* (n = 110), and *Bifidobacterium* (n = 98) (Fig. 1f). In addition to these abundant taxonomies in the human gut, some phages were predicted to infect *Klebsiella* (n = 8), *Akkermansia* (n = 6) and *Eggerthella* (n = 5), which are relatively minor in abundance but known to be associated with human health and disease[32–34] (Supplementary Data 6). Of the vOTUs whose hosts were predicted (n = 852), the majority (71.1%) were predicted to infect only one genus (i.e. specialist phages), whereas the others (28.9%) were predicted to infect multiple genera (i.e. generalist phages) (Fig. 1d).

Comparison of the phages across predicted hosts revealed that phages for Bacteroidetes phylum (e.g. *Bacteroides*, *Prevotella*, and *Parabacteroides*) had relatively larger genomes (median size: 54.4–65.0 kb), whereas those for Actinobacteria (e.g. *Bifidobacterium* and *Collinsella*) had smaller genomes (18.8 kb and 28.1 kb, respectively) (Fig. 1f), consistent with variations in the hosts' genome size[35,36]. In addition, the proportion of specialist and generalist phages differed substantially across the genera, in which the majority of phages predicted to infect *Bifidobacterium*, *Streptococcus*, and *Faecalibacterium* were specialist phages (87.3–98.8%) while those for *Clostridium*, *Roseburia*, and *Eubacterium* were likely to be generalist phages (83.3–97.6%) (Fig. 1f). The proportion of virulent and temperate phages also varied among the hosts; most of the phages for *Odoribacter*, *Bacteroides*, and *Parabacteroides* were predicted as virulent (76.1–88.9%), whereas those for *Roseburia*, *Dorea*, and *Anaerostipes* were temperate (83.3–91.3%).

To clarify the more distant relationship of the vOTUs, we further clustered vOTUs based on the proportion of shared proteins (>20%), providing a total of 223 viral clusters (VCs) corresponding to the family or subfamily level[7,37] (Supplementary Fig. 2b). Rarefaction analysis showed that the number of VCs, but not the number of vOTUs, was saturated at the number of individuals in this study (n = 4198) (Fig. 1e).

### Identification of novel phage clades abundant in the human gut

Owing to the lack of a high-quality genome catalogue of human gut phages until recently, the major clades of phages in the human gut are still largely unknown, other than a few groups, such as crAss-like phages[5,7,8,10,38,39]. To explore the major and abundant phage clades in the human gut, we quantified the abundance of each vOTU/VC by mapping the metagenomic reads of the 4198 individuals to the catalogue (**Methods**). On average, 1.8 ± 0.013% (mean ± s.e.) of the whole metagenomic reads were mapped to the catalogue, and 115 ± 0.62 vOTUs and 47 ± 0.17 VCs (mean ± s.e.) were detected from an

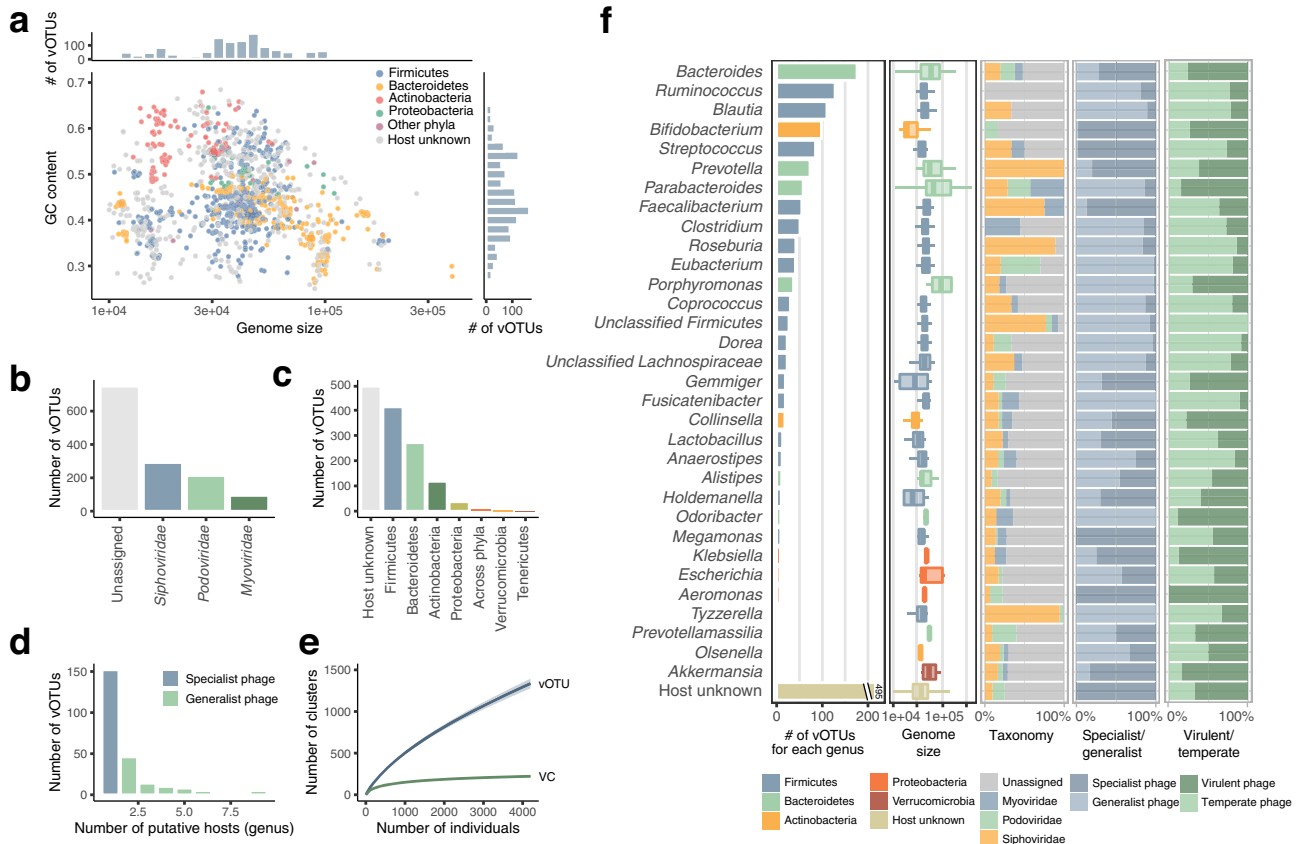

**Fig. 1 | Overview of reconstructed phage genomes from 4198 human gut metagenomes. a** Genome size and GC content of phage genomes (*n* = 4709) reconstructed from the dataset of 4,198 whole metagenomes. Bar plots on the top and right side depict the distribution of genome size and GC content, respectively. **b** Taxonomic annotation of the vOTUs at the family level. **c** Predicted hosts of the vOTUs at the phylum level. **d** Number of predicted hosts at the genus level for each vOTU. **e** Rarefaction curves of the detected vOTUs and VCs (mean number) in this cohort (*n* = 4,198). Shadows of the lines represent 95% confidence intervals.

**f** Number of vOTUs, genome size, taxonomy, the ratio of specialist and generalist phages, and the ratio of virulent and temperate phages for each predicted host (at the genus level). If a vOTU was predicted to infect more than one genus, it was distributed to all the predicted genera. Thirty-three genera predicted to be infected with more than 5 vOTUs are shown in the figure. In boxplots, boxes represent the interquartile range (IQR), and the lines inside show the median. Whiskers denote the lowest and highest values within 1.5 times the IQR.

individual. The accuracy of viral quantification based on whole metagenomic reads was confirmed by the comparison of viral profiles obtained from the VLPs metagenomes with those from whole metagenomes prepared from the same 24 fecal samples, which showed a high correlation between the two profiles (average Spearman correlation [$r_s$] = 0.74 and 0.76 at the vOTU and VC levels, respectively, Supplementary Fig. 4a). Additional cluster analysis showed clusters pairing each VLP with the whole metagenomes for the 24 samples (Supplementary Fig. 4b).

The most abundant VC in this cohort was a crAssphage-containing VC (VC_19) (relative abundance = 4.3 ± 0.17% [mean ± s.e.], Supplementary Note), which included crAssphage and crAss-like phages (candidate genera I, III, IV, and IX). Interestingly, we also identified several VCs whose abundance was at the same order of magnitude as VC_19 (Fig. 2a). Particularly, 9 VCs other than VC_19 were highly abundant (reads per kilobase million [RPKM] > 1) and prevalent (>100 genomes) across the 4198 whole gut metagenomes in this cohort. Of these abundant VCs, VC_6 was a group of crAss-like families (candidate genera VII, VIII, and X), and the phages in VC_1 showed similarities to the recently proposed new clade of Gubaphage/Flandersviridae[9,10]. The other seven abundant VCs (VC_2, 24, 12, 15, 3, 44 and 18) did not show any similarity to known phages in RefSeq, suggesting that they were novel phage clades abundant and prevalent in the human gut. They were detected in 24.6–90.2% of the individuals and were predicted to infect major gut species such as

*Bacteroides*, *Prevotella*, *Roseburia*, or *Bifidobacterium* (Supplementary Fig. 5, Supplementary Data 7). By searching genes specific to temperate phages[40] or comparing with reference microbial genomes in RefSeq, we found that the novel phage clades VC_2, 24, 12, 3 and 15 were likely to have lysogenic life cycles but the others (VC_44 and 18) were not (Supplementary Data 7). Phylogenetic analysis based on large terminases, portal proteins, and major capsid proteins revealed that most of the 7 VCs were monophyletic and significantly different from those of the known reference phages in RefSeq (Fig. 2c, Supplementary Fig. 6). In addition, a comparison of genomic similarities among the phages (**Methods**) revealed that most of the phages in the same VC were clustered together and distinct from the phages in the other VCs (Fig. 2b).

To investigate whether the VCs newly identified in this study exist as virus particles in the human gut, we explored them in the VLP dataset. We found that all of the 10 abundant VCs were detected in the 24 VLP dataset, and their relative abundance was, on average, 4.5–1115 times higher in the VLP metagenomes than in whole metagenomes prepared from the same faecal samples (Supplementary Fig. 7), clearly showing that they were present as virus particles in the gut. We named the seven novel VCs according to the names of the cities in which the research institutes of the authors are located (VC_2, "Toyamaviridae"; VC_24, "Konodaiviridae"; VC_12, "Shinjukuviridae"; VC_15, "Okuboviridae"; VC_3, "Tsurumiviridae"; VC_44, "Suehiroviridae"; and VC_18, "Umezonoviridae").

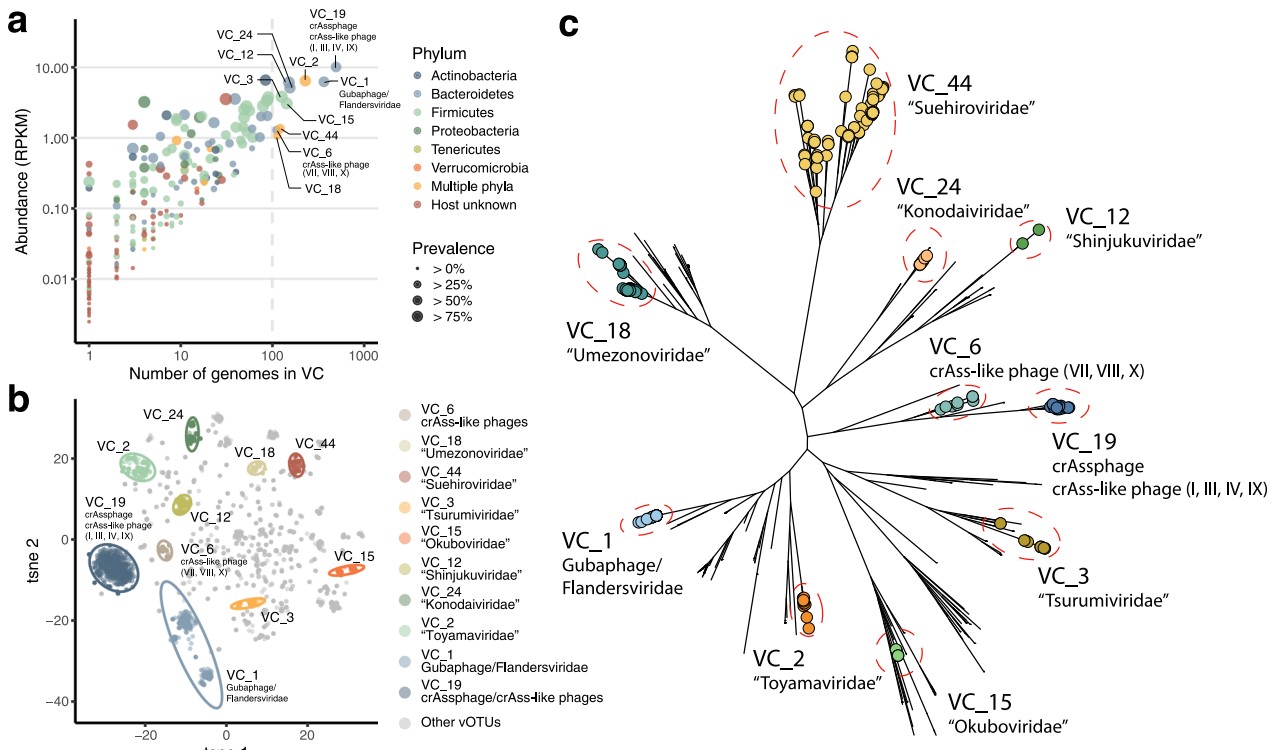

**Fig. 2 | Identification of novel viral clusters abundant and prevalent in the human gut. a** Average abundance of each VC among the 4198 individuals and the number of genomes forming the VC. The colour and size of each circle represent the phylum-level taxonomy of the predicted host and prevalence of the VC in the cohort, respectively. **b** Similarity of phage genomic content (proportion of shared proteins) visualised by tSNE. The colour of each circle represents the VC assigned to the genome. **c** Phylogenetic trees of the 10 most abundant and prevalent VCs in this cohort and phages in the RefSeq database were constructed based on large terminase proteins. Circles on the edges show vOTUs belonging to the VC and edges without circles represent reference genomes in RefSeq. Only representative genomes for each vOTU are included in the trees.

## Close interactions between the gut virome and bacteriome

It has been suggested that phages in the human gut largely affect the structure of the bacteriome through bacterial lysis and integration as prophages[1–3], but the relationships between phages and bacteria in the gut environment (i.e. *in natura*) are not well characterised. To explore this, we compared viral and bacterial profiles collected from the 4198 individuals and found significant positive correlations for α-diversity (Shannon diversity) and β-diversity (Bray-Curtis distance) between them ($r_s$ = 0.73 and 0.46, respectively; Fig. 3a, b). This results indicate that the virome and bacteriome structures are closely related to each other in the human gut. We also found that the β-diversity of the virome was significantly higher than that of the bacteriome (Fig. 3b, Supplementary Fig. 7), suggesting that the virome is more specific to each individual than the bacteriome.

To further explore their associations, we next examined one-to-one correlations between the relative abundance of each phage and its predicted host at the genus level among 4198 individuals (**Methods**). Notably, we found a positive correlation with an average $r_s$ of 0.18 between them (Fig. 3d), suggesting that phages and host bacterial species co-occur rather than being mutually exclusive in the human gut. Among the genera examined, *Megamonas*, *Escherichia*, *Prevotella* and *Lactobacillus* showed relatively higher correlation with their phages than other genera (Fig. 3c). These high correlations could be explained by the higher proportion of specialist phages that infect these genera as compared to other genera, such as *Clostridium*, *Ruminococcus*, and *Tyzzerella*, which are often infected by generalist phages (Fig. 3c). Indeed, the specialist phages showed a significantly higher correlation with their hosts ($r_s$ = 0.27 on average) than the generalist phages ($r_s$ = 0.08) (*P* < 2.2e-16, Fig. 3d). The higher correlation for specialist phages than generalist phages was reproduced with

other correlation indexes such as Pearson correlation and Maximal information coefficient, but was not present using Bray-Curtis dissimilarity (Supplementary Fig. 9a). Phage lifestyle did not have substantial impacts on the phage-host correlation, but virulent phages showed slightly but significantly stronger correlation ($r_s$ = 0.17) than temperate phages ($r_s$ = 0.15) (*P* = 0.025, Supplementary Fig. 9b).

Among the diverse genes encoded in the human gut bacteriome, the CRISPR-Cas, restriction-modification (RM), and abortive infection (Abi) systems are well-known defence mechanisms of prokaryotic species protecting against mobile genetic elements, including phage infection[41]. To explore associations between such antiviral genes and phages in the community, we quantified these genes and assessed their associations with the virome structure (**Methods**, Supplementary Data 8). We found that Shannon diversity of the virome was significantly higher in samples with abundant defence mechanisms than in those with low abundances of all three systems (Fig. 3e). In addition, significant positive associations were also identified for various subtypes in the CRISPR-Cas and RM systems (Fig. 3f). Other genes such as integrase and spore germination proteins, the latter of which is associated with species in Firmicutes[42], were also positively correlated with virome diversity (Supplementary Data 9).

## Comprehensive identification of host and environmental factors associated with the virome

To investigate how the gut virome structure is associated with host physiologies and environmental factors, we conducted an association analysis with age and sex, which are strong determinants of the gut bacteriome structure[43]. Age showed a significant positive correlation with virome diversity ($r_s$ = 0.20, *P*-value <2.2e−16, Fig. 4a), consistent with a positive correlation between age and bacteriome diversity

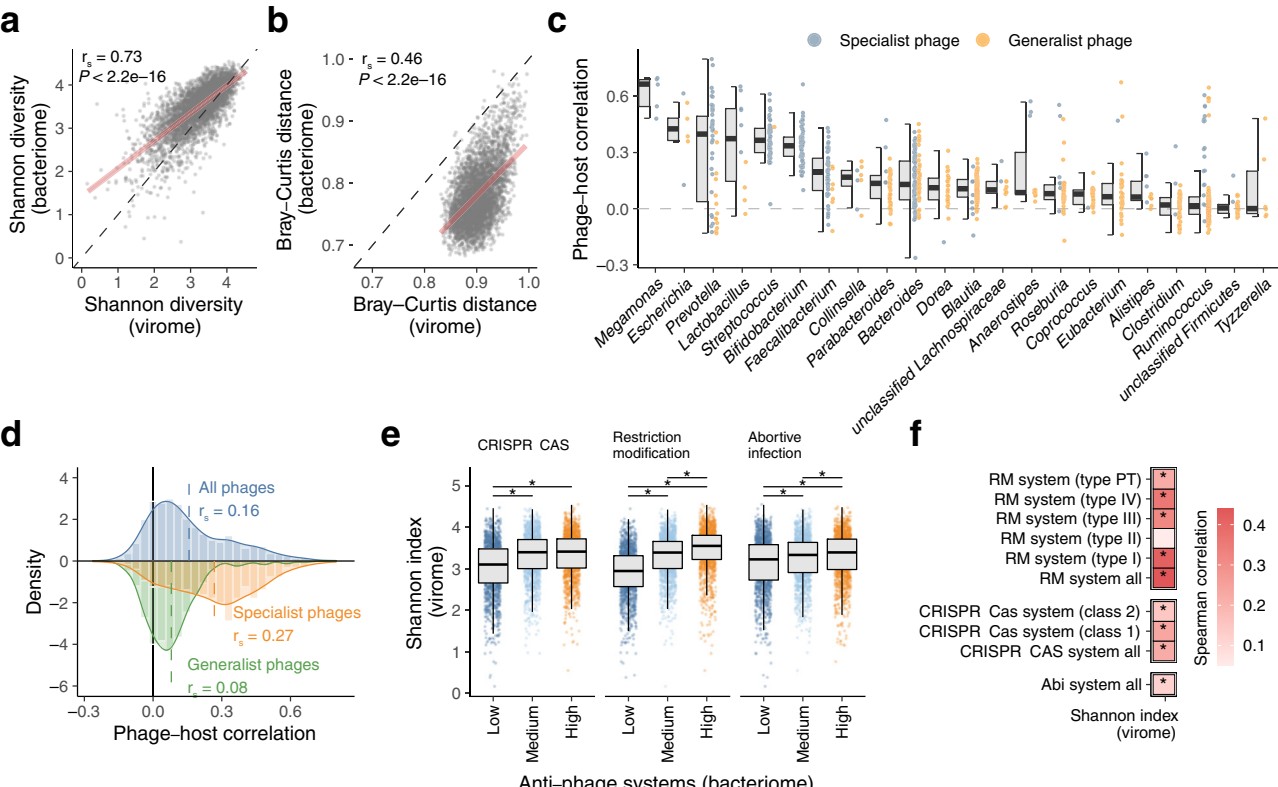

**Fig. 3 | Close interactions between the gut virome and bacteriome.** Comparisons of α-diversity (Shannon index) (**a**) and average β-diversity (Bray-Curtis distance) (**b**) between the virome and bacteriome of the human gut (*n* = 4,198 individuals). In **b** each circle represents the average value of the Bray-Curtis distance against other individuals. Regression lines are indicated in red. vOTU level and species level profiles of the virome and bacteriome were used, respectively. **c** Spearman correlations between relative abundances of vOTUs and predicted hosts at the genus level. Blue and orange circles represent specialist phages (i.e. phages predicted to infect only one genus) and generalist phages (i.e. phages predicted to infect more than one genus), respectively. Twenty-two genera (average relative abundance > 0.1%) with at least 5 vOTUs are shown in the figure. **d** Distribution of phage-host correlations across the 4,198 individuals. Blue, orange, and green colours represent the distributions of all phages, specialist phages, and generalist phages, respectively. Dashed lines show the average correlation in the distribution. **e** Comparison of the Shannon index of the virome among groups of individuals with high, medium, and low abundances of defence genes in the metagenomic data. The 4,198 individuals were placed into the three groups based on tertiles of the total abundances for the anti-phage systems. Asterisks represent statistical significance (*P* < 0.05, Wilcoxon rank-sum test, two-sided). **f** Heatmap summarising Spearman correlations between relative abundance of the anti-phage systems and the Shannon index of the virome. Asterisks represent statistical significance (*P* < 0.05). In boxplots, boxes represent the interquartile range (IQR), and the lines inside show the median. Whiskers denote the lowest and highest values within 1.5 times the IQR. Spearman correlation and its statistical significant were calculated using the cor.test function in R (two-sided).

(Supplementary Fig. 8a). At the level of the host bacteria, age had significant positive correlations with Proteobacteria phages and host unknown phages, but a negative correlation with Actinobacteria phages ($r_s$ = 0.10, 0.12 and −0.07, respectively, *P*-values <0.05, Fig. 4b), also in agreement with age-related changes in the bacteriome structure (Supplementary Fig. 8b, d). Multivariable analysis considering the effects of other covariates (**Methods**) revealed that age had significant associations with 176 vOTUs (*P*-values <0.05, Fig. 4c, Supplementary Data 10), including positive correlations with phages for *Clostridium*, *Ruminococcus* and *Faecalibacterium*, as well as host-unknown phages. At the VC level, age showed a strong negative correlation with *Bifidobacterium*-related VC_28, which was the most abundant VC among individuals in their 20 s, but decreased substantially with age (Supplementary Fig. 8c, Supplementary Data 10). Sex showed significant associations with 68 vOTUs and 24 VCs (Fig. 4d, Supplementary Data 10 and 11). Males had significantly higher abundances of phages predicted to infect *Prevotella* and *Megamonas*, while females had higher abundances of *Faecalibacterium*- and *Ruminococcus*-related phages, possibly reflecting differences in the bacteriome between males and females (Supplementary Fig. 8e).

To further explore the comprehensive relationships, we next performed an association analysis between the viral profiles and 232 host/environmental factors exhaustively collected from the 4198

individuals (Supplementary Data 2). Redundancy analysis showed that these factors explained 0.6% of the total variance in the virome at the vOTU level (Fig. 5a), which was substantially lower than the value that the same factors explained in the gut bacteriome at the species and genus levels (4.9% and 10.0%, respectively; Fig. 5a). The metadata categories most strongly associated with gut virome variation were clinical factors, such as medication and disease (explained variance = 0.5% and 0.3%, respectively). Permutational analysis of variance revealed that 97 of the 232 factors were significantly associated with virome variation (false discovery rate [FDR] < 0.05), among which age showed the strongest association with the virome (Fig. 5b, d). The significantly associated factors included various diseases (e.g. HIV infection, inflammatory bowel disease, past history of gastrointestinal resection), medications (e.g. osmotically acting laxatives, antiviral drugs, and alpha-glucosidase inhibitors), and diets (e.g. fruits, dairy products, and milk), which included numerous factors not identified previously[14,44]. The variation explained by each factor was highly correlated between the virome and bacteriome ($r_s$ = 0.87, *P*-value <2.2e−16, Fig. 5c). The vOTU of crAssphage (vOTU_974) showed no significant association with any metadata in this cohort, suggesting the presence of still unknown host, environmental or ecological factors that explain the variation of this most abundant phage in the human gut.

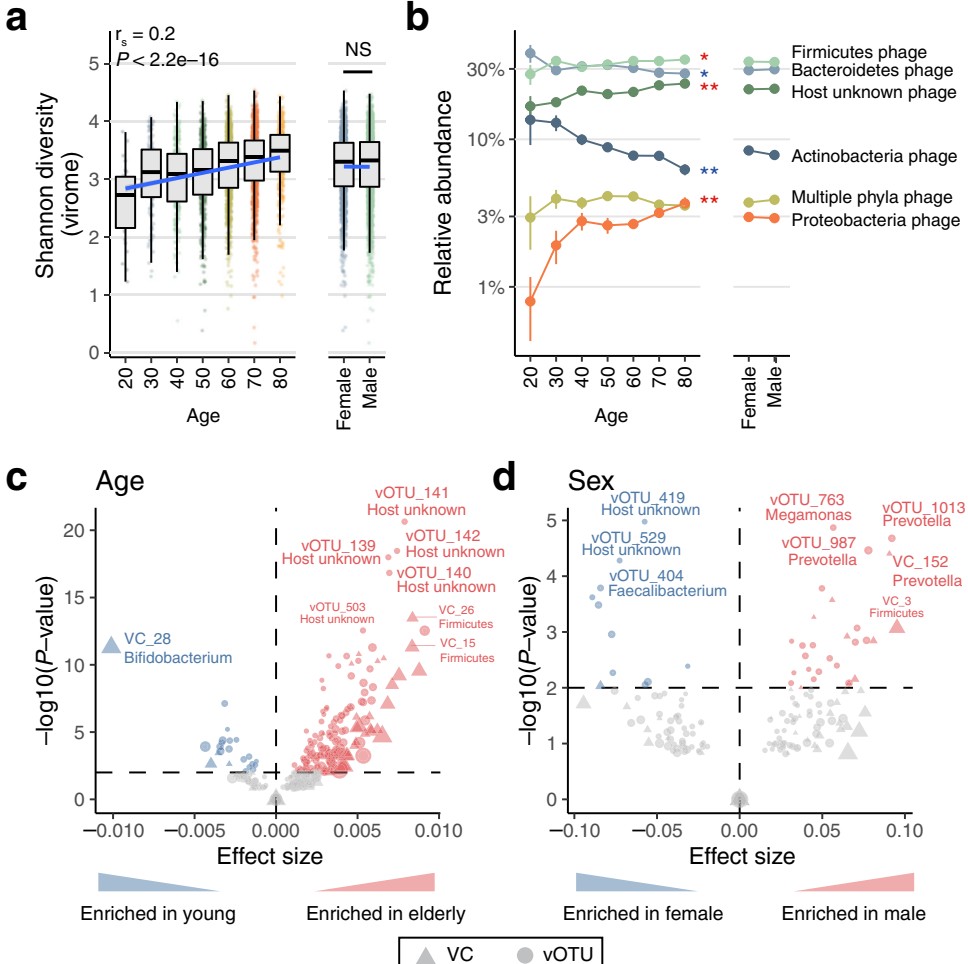

**Fig. 4 | Age- and sex-related changes in the human gut virome. a** Correlation between age/sex and the Shannon index of the virome. Individuals younger than 20 years (n = 2) and older than 80 years (n = 6) were excluded from the plot due to the low numbers of individuals. Spearman correlation and its statistical significant was calculated using the cor.test function in R (two-sided). **b** Average relative abundance of phages grouped according to the phylum-level taxonomy of host bacterial species. Each plot represents the average (mean) relative abundance of all the phages predicted to infect the phylum at each age. Red and blue asterisks show significant positive and negative correlations, respectively

(*FDR < 0.01, **FDR < 0.001). Error bars represent standard erros. **c, d** Associations of the viral profile with age (**c**) and sex (**d**). X- and Y-axes show the effect size and $\log_{10}$-transformed P-values obtained from multivariable regression analysis, respectively. The circles and triangles in the plots represent vOTUs and VCs, respectively. No correction for multiple testing was performed here since only variables with FDR < 0.05 in the univariate regression analysis were included in the multivariable regression analysis (**Methods**). In boxplots, boxes represent the interquartile range (IQR), and the lines inside show the median. Whiskers denote the lowest and highest values within 1.5 times the IQR.

Among the 97 factors with statistical significance, the majority showed strong associations with both the virome and bacteriome (Fig. 5d). By contrast, proton pump inhibitors, which showed the strongest effect on the gut bacteriome, had a relatively moderate effect on the virome (Fig. 5d). A large number of bacterial species in the oral cavity reach the gut following the administration of proton pump inhibitors, but these species are less transcriptionally active[45] and might have fewer interactions with gut phages[46]. Furthermore, antimicrobial drugs, such as cephalosporins, macrolides, and sulphonamides, which had large effects on the bacteriome, also had a moderate effect on the virome (Supplementary Fig. 8f), which might be explained by the absence of phage-originated molecular targets for anti-microbial drugs. Thus, ecological and biological differences between the virome and bacteriome might result in differences in the strengths of associations with some host and environmental factors.

## Discussion

In the present study, we have performed a large-scale analysis for viral profiles of deeply phenotyped individuals (n = 4198) and shown extensive virome variation and its association with the bacteriome and

numerous host and environmental factors. This study is, to the best of our knowledge, the largest single cohort analysis for the human gut virome with little biases due to the DNA amplification. The analysis uncovered novel viral clades, interactions between the virome and bacterial anti-viral genes, and clinical factors strongly associated with the virome, expanding our knowledge of the gut virome structure and variation.

We uncovered a lot of prevalent but previously uncharacterized dsDNA phage clades in the human gut (Fig. 2). Although recent large-scale studies have been uncovering numerous phage genomes in the human gut[9,10,15,31], our result suggests that the human gut phages are still under-explored and further efforts are needed to construct a more comprehensive phage catalogue of the gut phages. Some of the novel clades identified in this study could infect Firmicutes and Actinobacteria (VC_2, 15, 3, 44, and 18), which is in contrast to recently proposed large clades, such as crAssphage, crAss-like phage, and Gubaphage/Flandersviridae, all of which are suggested to infect Bacteroidetes[6,7,9,10]. Firmicutes is one of the major taxa in the human gut and include clinically important species[47]. Actinobacteria, such as *Bifidobacterium*, includes species used for probiotics and are quite

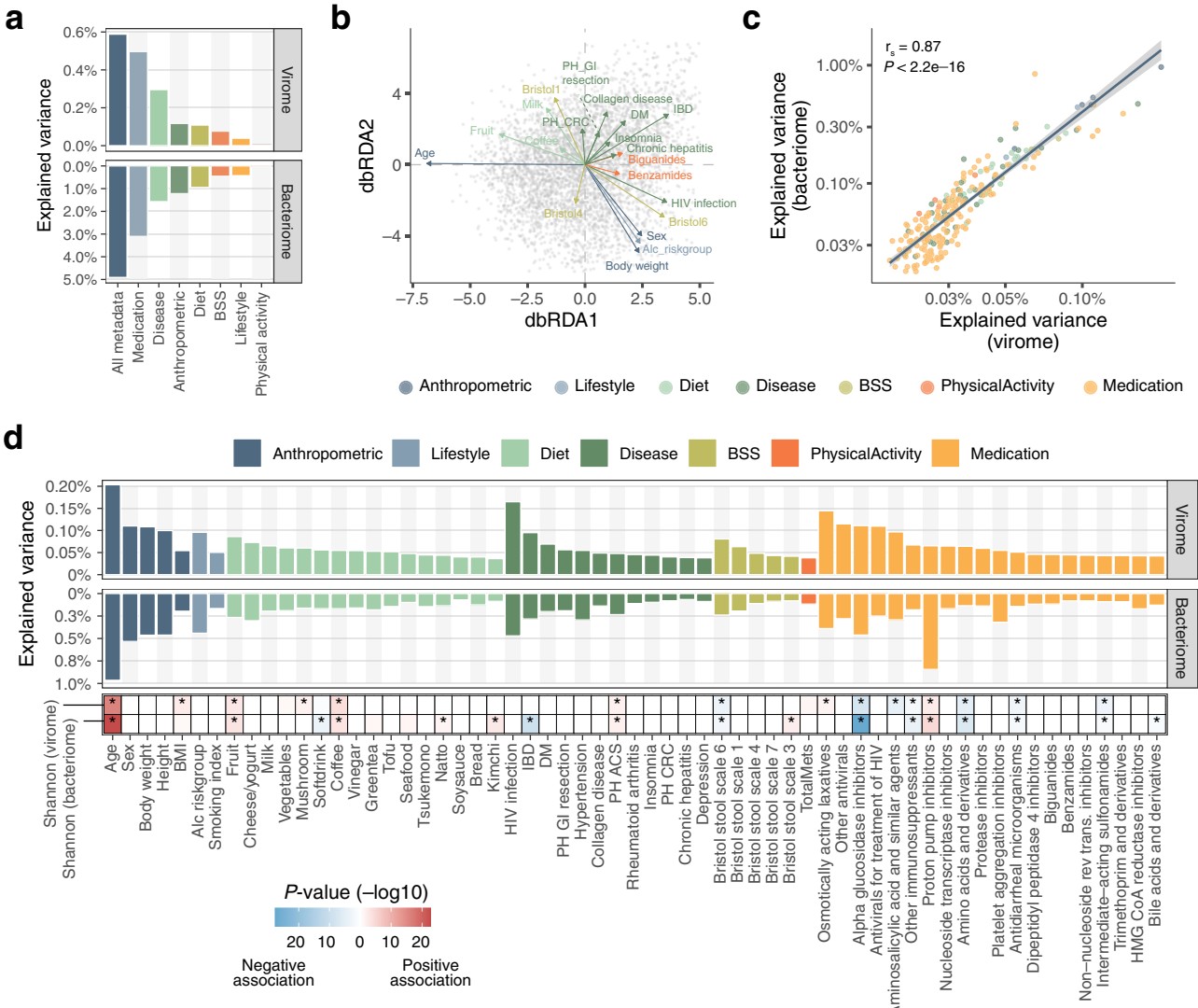

**Fig. 5 | Host and environmental factors significantly associated with the gut virome. a** Proportion of the variance explained by each metadata category in the gut virome and bacteriome ($n$ = 4,198 individuals) assessed by redundancy analysis. **b** Distance-based redundancy analysis showing the association between the 4,198 virome profiles at the vOTU level and collected metadata. Gray plots show each viral profile, and arrows represent associations with the metadata. The length of the arrow indicates the strength of the association, and the colour represents the metadata category. The top 20 non-redundant metadata with the strongest associations are shown in the plot. **c** Positive correlations between the explained variance of the gut virome and bacteriome by each metadata. Each point represents metadata and its colour indicates the metadata category. The blue line represents the regression line, and the grey shadow shows 95% confidence interval. Spearman correlation and its statistical significant was calculated using the cor.test function in

R (two-sided). **d** Proportion of the explained variance in the gut virome and bacteriome by each metadata assessed by permutational analysis of variance (FDR < 0.05). The bottom heatmap shows adjusted $P$-values for associations between the metadata and the virome diversity assessed by multivariable regression analysis adjusting for other covariates. Red and blue colors represent positive and negative associations, respectively. No correction for multiple testing was performed here since only variables with FDR < 0.05 in the univariate regression analysis were included in the multivariable regression analysis (**Methods**). For medication, the top 20 drugs with the strongest associations with the virome are shown. Abbreviations: BSS, Bristol stool scale; DM, diabetes mellitus; IBD, inflammatory bowel disease; CRC, colorectal cancer; ACS, acute coronary syndrome; PH, past history; GI, gastrointestinal tract.

abundant in the guts of infants[48]. The novel phage clades identified in this study may play important roles in the bacteriome by interacting with these major bacterial species and regulating their population dynamics.

Comparative analysis among the 4198 individuals showed a significant positive correlation between the diversity of virome and bacteriome (Fig. 3a), consistent with previous studies[16,49]. Interestingly, at the level of individual phages and hosts, their relative abundances were also positively correlated and they co-occurred (Fig. 3c). Since phages kill their hosts, a negative correlation between phages and their hosts would be expected. Actually, previous studies observed negative correlations between them in a time-series dataset using a mouse

model[50,51]. At the same time, however, phages have a limited host range and cannot live without their hosts[52,53] (Supplementary Data 5), suggesting that their symbiotic relationship results in a positive correlation at a more global scale (e.g. among different individuals and environments). Thus, our findings strongly suggest that, at the population level, the distribution of host bacteria is a dominant factor governing the distribution of individual phages in the human gut. Furthermore, we found that prokaryotic defence systems, such as the CRISPR-Cas, RM, and Abi systems, had significant associations with virome diversity (Fig. 3d). These associations suggest that anti-phage systems are important factors in shaping the virome through regulation of phage infection. Alternatively, given that numerous gut

microbes are infected and lysed by phages[51], the ability to protect against phage infection may also have a role in determining the structure and diversity of the bacteriome. However, our results based on the cross-sectional design cannot rule out the possibility that the defence systems were simply correlated with other microbiome properties (e.g. other gene functions, microbial density, and spatial structure) that actually cause the increase in virome diversity.

Our association analysis revealed numerous intrinsic/extrinsic factors significantly associated with the gut virome, which included previously unidentified factors, especially in disease and medication (Fig. 5). Strong associations with the virome were observed for host age, disease, and medication, in accordance with associations between these factors and the bacteriome (Fig. 4b). Additionally, the predicted hosts of the associated phages were mostly consistent with the microbial species associated with the factor (Figs. 4 and 5, Supplementary Fig. 10), supporting strong interactions between phages and their bacterial hosts. However, the variation in the virome explained by these factors was only 0.6%, significantly lower than the variations in the bacteriome (5%–10%) that was explained by factors in this cohort (Fig. 5a) as well as other cohorts[54,55]. This was possibly owing to substantially higher inter-individual variation in the virome compared to that of the bacteriome (Fig. 3b), which might be driven by various viral and bacterial ecological factors, such as the rapid evolutionary rate of phages[56], strain-level diversity of bacterial species among individuals[57], or the associated variability of the defence systems[58]. More detailed analysis and understanding of inter-individual diversity of the virome is needed since its variation has potential clinical impacts on faecal microbiota transplantation and phage therapy[59].

In summary, our population-level analysis of the human gut virome uncovered its substantial variation and associations with the corresponding bacteriome and various factors. These results provide the basis for a better understanding of viral and microbial ecology in the human gut and are anticipated to facilitate medical and industrial applications to the gut microbial community.

## Methods

### Sample collection and metagenomic sequencing

Written informed consent was obtained prior to participation in the project. The study protocol for the Japanese (Disease, Drug, Diet, Daily life) microbiome project was approved by the medical ethics committees of the Tokyo Medical University (Approval No: T2019-0119), National Center for Global Health and Medicine (Approval No: 1690), the University of Tokyo (Approval No: 2019185NI), Waseda University (Approval No: 2018-318), and the RIKEN Center for Integrative Medical Sciences (Approval No: H30-7). We conducted a prospective cross-sectional study of 4198 individuals participating in the Japanese 4D microbiome project, which commenced in January 2015 and is ongoing[20].

Participants registered in the project were those who visited hospitals in the area for disease diagnosis or a health checkup. Faecal samples are collected from both healthy and diseased participants. The eligibility criteria for participants are as follows: (1) born and raised in Japan; (2) age >15 years; (3) written informed consent provided; and (4) having an endoscopic diagnosis on colonoscopy; either having undergone a colonoscopy within the last 3 years or planning to undergo colonoscopy for colorectal cancer screening, surveillance, and diagnosis of various gastrointestinal symptoms. The exclusion criteria were as follows: (1) suspected acute infectious disease based on clinical findings (e.g., acute enterocolitis, pneumonia, tuberculosis etc.); (2) acute bleeding; (3) hearing loss; (4) unable to understand written documents; (5) unable to write and (6) limited ability to perform activities of daily living. No compensation was paid to participants.

Participants collected faecal samples using a Cary–Blair medium-containing tube[60] at home, and the samples were refrigerated for up to 2 days before the hospital visit. Immediately after participants arrived at the hospital, their faecal samples were frozen at −80 °C until DNA extraction. We avoided collecting samples within 1 month of administering bowel preparation for colonoscopy because it has a profound effect on the gut microbiome and metabolome[61]. Health professionals checked that the amount of stool was sufficient for analysis. Shotgun metagenomic sequencing was performed for 4241 faecal samples and quality controls were conducted[20], from which 43 samples were excluded from further analyses due to the low number of high-quality reads (<5 million reads) as described in detail previously[20] (Supplementary Data 1). To explore the viral profiles of VLPs and whole metagenomes from the same samples, we collected additional faecal samples from 24 individuals in the same manner as described earlier.

### Metadata collection

Details for metadata collection were described previously[20]. Briefly, the participants completed a self-reported questionnaire on body weight, height, alcohol consumption, smoking, dietary habits, physical activity, and Bristol Stool Scale score[62]. Health professionals checked the entries to correct obvious inaccuracies and obtain any missing data. BMI was categorised into five groups according to the standard World Health Organization (WHO) classification and considering the threshold value of mortality risk[63] (0, underweight, <18.5 kg/m²; 1, normal weight [low], 18.5–20.0 kg/m²; 2, normal weight [high], 20.1–24.9 kg/m²; 3, overweight, 25.0–29.9 kg/m²; and 4, obese [≥ 30.0 kg/m²]). Dietary habits were assessed using a 7-point Likert scale (1, never or rarely; 2, 1–3 times/month; 3, 1–3 times/week; 4, 4–6 times/week; 5, 1 time/day; 6, 2 times/day; and 7, ≥3 times/day). Physical activity was evaluated with the International Physical Activity Questionnaire–Short Form[64]. Exercise reported as vigorous intensity, moderate intensity, or walking was denoted as 1, and <60 min/week was denoted as 0. The total sitting hours per day and the total metabolic equivalent of tasks[65] were divided into four groups based on quartiles for the entire dataset. For the diagnosis of gastrointestinal diseases, an electronic high-resolution video endoscope was used. Comorbidities, or a history of hypertension, dyslipidemia, and any component of the Charlson comorbidity index[66], were evaluated. The definite diagnosis of the disease was based on histopathological or cytological examinations or imaging modalities (e.g. computed tomography, magnetic resonance imaging and ultrasound). For medication, health professionals evaluated entries in the participant's medication pocketbook (the *Okusuri-techo*) made by pharmacists when filling prescriptions[20] to ensure that there were no omissions or discrepancies with the self-reported data. Electronic medical records were also checked to identify medications used. Drug use was defined as oral or self-injected administration within the previous month. All medications with pharmaceutical brand names were grouped according to the WHO's Anatomical Therapeutic Chemical classification system (4th level)[67]. In total, 232 metadata were assessed and used in this study.

### Preparation of VLP DNA and sequencing

Frozen faecal samples (30–500 mg) were suspended in a 2.5 mL SM buffer with 0.01% gelatine by vortexing and centrifuged at 5000 × g for 10 min at 4 °C to remove debris. The supernatant was filtered with 5.0 μm and 0.45 μm PVDF pore membrane filters (Millex-HP Syringe Filter; Merck Millipore) to remove bacterial cells. An equal volume of 20% polyethylene glycol solution (PEG-6000-2.5 M NaCl) was added to the filtrate and stored overnight at 4 °C. The solution was centrifuged at 20,000 × g for 45 min at 4 °C, and the supernatant was discarded to collect VLPs. The VLP pellet was suspended in 1 mL SM buffer with lysozyme (10.0 mg/reaction; Sigma Aldrich) and incubated for 60 min at 37 °C with gentle shaking to degrade unfiltered bacterial cells. The lysate was incubated with 10 U DNase (NIPPON GENE), 5 U TURBO DNase (Thermo Fisher Scientific), 5 U Baseline-ZERO DNase (Epicentre), 25 U Benzonase (Sigma Aldrich), and RNase (25 g/sample;

NIPPON GENE) in DNase buffer (1× concentration) for 1 h at 37 °C with gentle shaking. To inactivate the DNases, EDTA (final concentration 20 mM) was added to the DNase-treated lysate and heated for 15 min at 70 °C. Proteinase K (0.5 mg/reaction; Sigma Aldrich) and SDS (final concentration 0.1%) were added to the VLPs and gently mixed at 55 °C for 30 min. An equal volume of phenol/chloroform/isoamyl alcohol (Life Technologies Japan, Ltd) was added to the lysate and gently mixed for 10 min at room temperature (20–25 °C). The lysate was centrifuged at 9000 × g for 10 min at 25 °C, and the aqueous phase was collected. Sodium acetate (final concentration 0.3 M) and an equal volume of isopropanol with Dr. GenTLE precipitation carrier (Takara Bio) were added to the DNA solution and pelleted by centrifugation at 12,000 × g for 15 min at 4 °C. The DNA pellet was rinsed with 75% ethanol and dissolved in TE buffer (10 mM Tris-HCl, 10 mM EDTA). An equal volume of polyethylene glycol solution (20% PEG6000-2.5 M NaCl) was added and kept on ice for at least 10 min, and the DNA was pelleted by centrifugation at 12,000 × g for 10 min at 4 °C. Finally, the DNA was rinsed with 75% ethanol, dried, and dissolved in TE buffer. For NovaSeq shotgun metagenomic sequencing, libraries were constructed from 2.5 ng VLP DNA using a KAPA HyperPrep Kit (KAPA Biosystems) with 12 cycles of amplification. The libraries were subjected to 150-bp paired-end sequencing on a NovaSeq platform.

Whole metagenomic DNA was also prepared from the same faecal samples (10 to 250 mg faeces) with an enzymatic lysis method as described previously[68]. Libraries were constructed from 100 ng whole metagenomic DNA and sequenced by NovaSeq using the same method as for VLP DNA.

### Identification of phages in the metagenomic data

To construct a high-quality double-stranded DNA (dsDNA) phage catalogue with minimum contamination of bacterial chromosome and plasmid sequences, we developed a custom pipeline and applied it to the 4198 whole gut metagenomes as described below. The metagenomic reads of each individual were assembled into contigs using the MEGAHIT assembler (v1.2.9)[69]. The circularity of the assembled contigs (>10 kb) was assessed using the check_circularity.pl script, included in the sprai assembler package (https://sprai-doc.readthedocs.io/en/latest/index.html), by modifying the threshold for terminal redundancy as follows: >97% identity and >130 bp. Encoded genes in the contigs were predicted by MetaGeneMark (3.38)[70]. Assembled contigs were defined as phages if they passed all of the following six criteria.

1. A genome size threshold was applied, and contigs less than 10 Kb were excluded, as typical dsDNA phages have genomes larger than >10 Kb[71].
2. Viral-specific k-mer patterns were checked by DeepVirFinder (v1.0)[22]. Contigs with p-values >0.05 were excluded from further analysis.
3. To detect viral hallmark genes (VHGs) and plasmid hallmark genes, we performed a highly sensitive HMM-HMM search against the Pfam database[72]. First, the encoded genes were aligned to the viral protein database, collected from complete (circular) viral genomes (n = 13,628) in the IMG/VR v2 database[30] using JackHM-MER. The obtained HMM profiles were searched against the Pfam database using hhblits[73] with a >95% probability cut-off. These procedures were performed using the pipeline_for_high_sensitive_domain_search script (https://github.com/yosuken/pipeline_for_high_sensitive_domain_search)[74,75]. Contigs with plasmid hallmark genes or those without VHGs were excluded. The hallmark genes used in this analysis are summarised in Supplementary Data 3.
4. The presence of housekeeping marker genes of prokaryotic species was checked by fetchMG (v1.0)[76], and ribosomal RNA genes (5 S, 16 S and 23 S) were identified by barrnap (0.9) (https://github.com/tseemann/barrnap). Contigs with the marker genes and ribosomal RNA genes were excluded from further analysis.
5. The encoded genes of each contig were aligned to the viral protein database and a plasmid protein database constructed from the reference plasmids in RefSeq (n = 16,136, in April 2020) using DIAMOND (v0.9.29.130)[77] with the more-sensitive option. The number of genes aligned to each database was compared, and contigs with more genes aligned to the plasmid protein database were excluded from further analysis.
6. The proportion of provirus regions was assessed by CheckV (v0.7)[24], and contigs estimated with <80% of provirus regions were excluded.

First, we screened complete phage genomes from the circular contigs using these six criteria (Supplementary Fig. 1a). To identify phage genomes that were not complete but were of high or medium quality, we next screened possible phage contigs in the linear contigs. We aligned genes identified in the linear contigs to gene sets obtained from the complete phage genomes identified in this study (n = 1125) and the IMG/VR database (n = 13,628). The alignment was performed using DIAMOND with the more-sensitive option and e-value <1E-5 as a threshold. Contigs were defined as possible phage contigs if >40% of the genes were aligned to genes from a complete phage genome and the size of the contig was >70% and <120% of the complete genome. For these possible phage contigs, the above six criteria were applied, and those that did not pass were excluded. Finally, CheckV was used to screen for excess host bacterial genomes and exclude linear contigs defined as low quality or having >10% contamination.

To evaluate the performance of this custom pipeline, we applied the pipeline to reference phage genomes (n = 2609, as positive data) and plasmid sequences (n = 16,136, as negative data) in Refseq. The true positive rate was defined as the number of phages detected as phages by the pipeline divided by the number of reference phages. The false positive rate was defined as the number of plasmids detected as phages by the pipeline divided by the number of reference plasmids. DeepVirFinder[22], VirSorter (v1.0.3)[23] Virsorter2 (2.2.3)[25], VIBRANT (v1.2.1)[26], Seeker (v1.0.3)[27] and ViralVerify (v1.1)[28] were also applied to the same datasets with the default parameters, and the performance was compared among them.

### Analysis of phage genomes

Viral operational taxonomic units (vOTUs) were constructed by clustering phage genomes with a > 95% identity[29] using dRep (v2.2.3)[78] with the default options. Representative sequences of each vOTU selected by dRep were further clustered with reference sequences in RefSeq, IMG/VR[30], gut virome database (GVD)[15], gut phage database (GPD)[9], and metagenomic gut virus (MGV) database[31] with >95% identity and >85% length coverage using aniclust.py script in the CheckV package to identify common sequences among the databases.

To further construct broader viral clusters (VC), proportions of protein clusters shared between phages were assessed. First, to define protein clusters, similarity searches of all protein sequences from all the phages identified in this study were performed using DIAMOND with the more-sensitive option (e-value <1E-5). Based on the similarities between proteins, protein clusters were defined by MCL (v14-137)[79] with an inflation factor of 2. The percentage of shared protein clusters was calculated for each phage pair, and phages sharing >20% of clusters were grouped as a VC, which corresponds approximately to family- or subfamily-level clusters[7,37]. Rarefaction curves of the vOTUs and VCs were estimated with the iNEXT function in the iNEXT package (v2.0.20)[80]. The similarity matrix of the phages based on the percentage of shared protein clusters was further projected by tSNE using the tsne function in the Rtsne package (v0.16).

Taxonomy annotation of phages was performed with a voting approach described previously[16] with minor modifications. First, the protein sequences of each phage were aligned to viral proteins detected from phage genomes in RefSeq (n = 2609, in April 2020)

using DIAMOND with the more-sensitive option. Then, the best-hit taxonomy of each protein (family levels) was counted, and the most common taxonomy was assigned to the phage if >20% of proteins in the phage were aligned to the same taxonomy.

Phage lifestyles (i.e. virulent or temperate) were predicted by BACPHLIP[40] and alignments to reference bacterial genomes in the RefSeq. Phages were defined as temperate if the BACPHLIP score was >0.8 or the phage genome was aligned to any reference genomes with >1000 bp alignment length with >95% identity.

## Host prediction

Bacterial and archaeal genomes were downloaded from the RefSeq database (in April 2019). To reduce the redundancy of genomes from closely related strains in the same species (e.g. *Escherichia coli*), 10 genomes were selected randomly for species with more than 10 genomes, and other genomes were excluded from the dataset. The reference dataset consisted of 33,215 bacterial and 822 archaeal genomes.

Host prediction of the identified phages was performed using CRISPR spacers[81]. CRISPR spacers were predicted from the reference microbial genomes and assembled contigs (>10,000 bp) from the 4198 metagenomic datasets using PILER-CR (1.06)[82]. Short (<25 bp) or long (>100 bp) spacers were discarded. In total, 679,323 and 283,619 spacers were identified from the reference microbial genomes and assembled contigs, respectively. Taxonomy information was assigned to the assembled contigs if they were aligned to the microbial reference genomes with >90% identity and >70% length coverage thresholds using MiniMap2[83]. The CRISPR spacers were mapped to the phage genomes using BLASTN with the option for short sequences: -a20 -m9 -e1 -G10 -E2 -q1 -W7 -F F[81]. CRISPR spacers, which were mapped with 100% identity or 1 mismatch/indel with >95% sequence alignment, were used for host assignment at the genus level. Assignments of host species were checked manually, and if any of the following non-human intestinal species were assigned, the host was excluded: *Dickeya, Anaerobutyricum, Rubellimicrobium, Eisenbergiella, Harryflintia, Leucothrix, Photorhabdus, Spirosoma, Syntrophobotulus, Thermincola, Algoriphagus, Franconibacter, Kandleria, Lawsonibacter, Methylomonas, Provencibacterium, Pseudoruminoccoccus, Rhodanobacter, Romboutsia, Sharpea, Varibaculum* and *Thioalkalivibrio*.

## Quantification of viral abundance and analysis of the virome profile

To quantify the viral abundances in each sample, metagenomic reads were mapped to the gene set of VHGs (Supplementary Data 3) of each representative vOTU using Bowtie2 with a > 95% identity threshold, and reads per kilobase million (RPKM) were calculated for each vOTU. The reason for using only VHGs in the analysis was to avoid over-counting of viral reads, which could be caused by spurious mapping of reads from horizontally transferred genes of other phages or bacterial species. The α-diversity (Shannon diversity) of the vOTU-level viral profile was calculated using the diversity function in the vegan package. The β-diversity (Bray-Curtis distance) between individuals was assessed using the vegdist function, and the average distance against other individuals was calculated for each individual. The VC-level viral profile was obtained by summing all the RPKM of vOTUs for each VC.

## Phylogenetic analysis of novel VCs

To construct phylogenetic trees for the vOTUs and reference genomes, protein sequences of large terminases, portal proteins, and major capsid proteins (Supplementary Data 3), which are often used to construct phage phylogenetic trees[7,9], were extracted from the vOTUs in the 10 most abundant VCs (VC_19, 1, 2, 24, 12, 15, 3, 44, 18, 6), and their homologues were searched for in the reference phage genomes in RefSeq using DIAMOND with the more-sensitive option (e-value <1E-5). The collected protein sequences were aligned by MAFFT (v7.458)[84]

with the linsi option, and the alignments were trimmed by Trimal (v1.4.rev15)[85] with the automated1 option. Phylogenetic trees were constructed by FastTree (2.1.10)[86]. The phylogenetic trees were visualised with iTOL (v5)[87]. For each VC, vOTUs with the highest number of genomes were selected, and their genomic structures were visualised by the circlize package (v0.4.15)[88].

## Taxonomic and functional analysis of the bacteriome

Taxonomic and functional profiles of the bacteriome were obtained as described previously[20]. Briefly, bacterial profiles at the species and genus levels were obtained with the single copy marker gene-based method using mOTUs (v2.1.1)[89]. Functional profiles at the Kyoto Encyclopaedia of Genes and Genomes (KEGG) orthology (KO) level were obtained by mapping the metagenomic reads to a non-redundant gene set constructed from the 4198 subjects' metagenomic data[20]. Functional annotation of the non-redundant genes was performed using eggNOG-mapper[90], in which DIAMOND was used for alignment to the eggNOG orthology database (version 4.5)[91].

KOs involved in prokaryotic defence mechanisms, such as CRISPR-Cas and RM (Supplementary Data 8)[58], were collected, and their total relative abundance in each system was calculated. Since functional annotation for the Abi system is not included in the KEGG database, we collected genes annotated as 'abortive phage infection' and 'abortive phage resistance' in the eggNOG annotation and calculated the total abundance. The 4198 individuals were classified into three groups (high, middle, and low) based on tertiles of the total abundance, and Shannon diversity of the virome was compared among the three groups by the Wilcoxon rank-sum test.

## Phage-host correlation analysis

To explore the phage-host association in the community, Spearman correlations between relative abundances of vOTUs and microbial species at the genus level were evaluated. If the vOTU was predicted to infect more than one genus (i.e. generalist phage), the correlation was calculated for every predicted host. If a phage-host pair was absent (0 abundance) in a sample, the sample was excluded from the correlation analysis. vOTUs with average relative abundance >0.01% (n = 865) and genera with average relative abundance >0.5% (n = 32) were included in the analysis.

## Analysis of VLPs and whole metagenomes from 24 faecal samples

Quality filtering of sequenced reads from the 24 VLPs and whole metagenomes was performed using fastp (version 0.20.1)[92] with the default parameters. Contamination with human (hg38) or phiX genomes was excluded by mapping the reads to the genomes using Bowtie2.

To exclude bacterial DNA contamination in the VLP dataset, we performed further filtering. First, the VLP reads were assembled into contigs using MEGAHIT and the contigs were checked for virus or not. Contigs were defined as viral contigs if they were predicted as viruses by DeepVirFinder (P-value <0.05) and did not encode rRNA and marker genes checked by barrnap and fetchMG, respectively. Then, VLP reads were mapped to the viral contigs using Bowtie2, and those not mapped to the viral contigs were excluded from the VLP dataset. Viral profiles at the vOTU and VC levels for the de-contaminated VLP and whole metagenomic datasets were obtained with the same methodology for the 4198-subject metagenomic dataset described earlier.

## Association analysis between the virome and various host/environmental factors

The association between each vOTU/VC and age/sex was assessed by multivariable regression analysis considering the effects of other covariates as described before[20]. Briefly, the relative abundance of each vOTU/VC was log$_{10}$-transformed, and single linear-regression

analysis was performed using the transformed abundance as a response variable and metadata as an explanatory variable. This single linear-regression analysis was performed for age, sex, and other metadata ($n = 230$, Supplementary Data 2), and metadata significantly associated with the vOTU/VC were determined (FDR < 0.05) by taking into account the total number of single regression analyses (number of vOTU/VCs multiplied by number of metadata). Then, multiple regression analysis was performed including all the significant metadata in the single regression analysis as explanatory variables. To exclude confounding factors, stepwise variable selection was performed based on Akaike's information criterion with the step function. Metadata was defined as significantly associated with the vOTU/VCs if they remained in the model with a $P$-value <0.05. All regression models were constructed using the glm2 function in the glm2 package (v1.2.1). In total, 390 vOTUs and 112 VCs, whose average relative abundances in the 4198 metagenomic dataset were >0.05% and >0.1%, respectively, were included in the analysis. For visualisation, individuals younger than 20 years ($n = 2$) and older than 80 years ($n = 6$) were excluded due to the low numbers of such individuals (Fig. 5a, b).

Stepwise redundancy analysis was performed to evaluate the total variance of the virome and bacteriome (relative abundance data) explained by each metadata category using the ordriR2step function in the vegan package (v2.5.7)[93]. To investigate the associations between the virome/bacteriome and each single metadata item, permutational analysis of variance was performed using the adonis function in the vegan package based on the Bray–Curtis distance with 10,000 permutations. $P$-values were corrected for multiple comparisons by the Benjamini–Hochberg method[94].

## Statistics
All statistical analyses were conducted using R (v3.5.0) with two-sided test and the Benjamini–Hochberg method for multiple comparisons unless otherwise stated. Of the 4211 metagenomic samples sequenced, 43 samples were excluded due to less reads (5 million) than the others. No statistical method was used to predetermine sample size.

## Reporting summary
Further information on research design is available in the Nature Research Reporting Summary linked to this article.

## Data availability
Sequence statistics of the 4198 individuals and cohort-level summaries of the metadata are available in Supplementary Data 1 and 2, respectively. All circular and linear phage genomes detected in this study ($n = 4709$) are available in the NCBI GenBank (PRJNA862966 [OP030729-OP031128 and OP072211-OP076519]) and at https://doi.org/10.5281/zenodo.5645361. Reference bacterial, archaeal, viral and plasmid genomes were downloaded from the RefSeq database. Genomes of human gut phage constructed in previous studies were downloaded as follows. GVD: https://datacommons.cyverse.org/browse/iplant/home/shared/iVirus/Gregory_and_Zablocki_GVD_Jul2020/. MGV: https://portal.nersc.gov/MGV/. GPD: http://ftp.ebi.ac.uk/pub/databases/metagenomics/genome_sets/gut_phage_database/. IMG/VR2: https://img.jgi.doe.gov/cgi-bin/vr/main.cgi.

## Code availability
The custom phage-detection pipeline used in this study is available at https://gitlab.com/suguru.nishijima/phage_detection.

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

## Acknowledgements

We thank K. Miki (National Center for Global Health and Medicine) for help with the data collection; M. Toya, S. Oshiro (National Center for Global Health and Medicine), and K. Oshima (The University of Tokyo) for assistance with the metagenomic data analysis; and W. Suda (RIKEN) for supporting VLPs sequencing. The super-computing resource was provided by the Human Genome Center (The University of Tokyo). This work was partially supported by a Grant-in-Aid for Early-Career Scientists (18K14682) from the Japan Society for the Promotion of Science to S.N.; Grants-in-Aid for Research from the National Center for Global Health and Medicine (28-2401, 29-2001, 29-2004, 19A1011, 19A1022, 19A2015 and 29-1025), the Ministry of Health, Labour and Welfare, Japan (grant number: 19HB1003), JSPS KAKENHI Grant (JP17K09365 and 20K08366), Smoking Research Foundation, DANONE RESEARCH GRANT, Research funding of Japan Dairy Association (J-Milk), Tokyo Medical University Cancer Research Foundation, Japan Agency for Medical Research and Development (AMED) (Research Program on HIV/AIDS: JP22fk0410051), and Takeda Science Foundation to N.N. The funding bodies had no role in the study design, data collection or analysis, decision to publish or preparation of the manuscript.

## Author contributions

S.N., N.N. and M.H. designed the study and wrote the manuscript. S.N. performed bioinformatic and statistical analyses. T.A. and Y. Kiguchi performed DNA extraction from faecal samples for whole-genome shotgun and VLP sequencing. N.N., K.M., and Y. Kojima. collected the metadata and integrated the drug information. M.K., M.O., K.U., S.O., M.M., T.I., T.K. and N.U. participated in the study's conception and edited the manuscript. M.H. supervised the study. N.N. is a principal investigator of the Japanese 4D microbiome project. All authors have read and approved the submitted version of the manuscript.

## Competing interests

The authors declare no competing interests.

## Additional information

[1]Graduate School of Advanced Science and Engineering, Waseda University, Tokyo, Japan. [2]Computational Bio Big Data Open Innovation Lab., National Institute of Advanced Industrial Science and Technology, Tokyo, Japan. [3]Department of Gastroenterological Endoscopy, Tokyo Medical University, Tokyo, Japan. [4]Department of Gastroenterology and Hepatology, National Center for Global Health and Medicine, Tokyo, Japan. [5]Laboratory for Microbiome Sciences, RIKEN Center for Integrative Medical Sciences, Yokohama, Japan. [6]Pathogenic Microbe Laboratory, Research Institute, National Center for Global Health and Medicine, Tokyo, Japan. [7]Department of Clinical Research Strategic Planning Center for Clinical Sciences, National Center for Global Health and Medicine, Tokyo, Japan. [8]Department of Diabetes, Endocrinology, and Metabolism, Center Hospital, National Center for Global Health and Medicine, Tokyo, Japan. [9]Diabetes and Metabolism Information Center, Diabetes Research Center, Research Institute, National Center for Global Health and Medicine, Tokyo, Japan. [10]Diabetes Research Center, Research Institute, National Center for Global Health and Medicine, Tokyo, Japan. [11]AIDS Clinical Center, National Center for Global Health and Medicine Hospital, Tokyo, Japan. [12]Genome Medical Sciences Project, Research Institute, National Center for Global Health and Medicine, Chiba, Japan. [13]Department of Gastroenterology and Hepatology, Tokyo Medical University, Tokyo, Japan. [14]Department of Gastroenterology and Hepatology, National Center for Global Health and Medicine, Kohnodai Hospital, Tokyo, Japan. [15]Present address: Structural and Computational Biology Unit, European Molecular Biology Laboratory, Heidelberg, Germany. [16]These authors contributed equally: Suguru Nishijima, Naoyoshi Nagata. ✉e-mail: nishijima.suguru@gmail.com; nnagata_ncgm@yahoo.co.jp

