## [Peer Review File · Nature Communications]

REVIEWER COMMENTS

Reviewer #1 (Remarks to the Author):

In their manuscript, “Population-level metagenomic exploration of gut virome variation”, the authors conduct an in depth analysis of the gut virome using 4,198 whole metagenomes of deeply phenotyped Japanese individuals. The authors developed their own bioinformatics pipeline to discover 1,125 complete and 3,584 draft viral genomes, which the authors estimate cover 57% of virome abundance based on mapping reads from VLP metagenomes. Clustering of the data revealed 1,347 viral operational taxonomic units, and these are explored for novel taxa. The authors used extensive phenotyping data to identify host and environmental factors that correlate with the gut virome. I found the study to be well conducted, well written, with excellent quality figures.

Major comments:

I have several comments and questions regarding the phage pipeline. First, it's unclear to me why a novel pipeline was needed for the current study, given the large number of recently published tools, including Virsorter 2, VIBRANT, viralVerify, and Cenote Taker 2, which all make use of viral marker genes, and several of which can handle provirus sequences. None of these tools are evaluated in the benchmark (extended data figure 1), whereas, I found it strange to benchmark and report accuracy for CheckV which is not a viral detection pipeline – this makes it seem as though existing tools are less accurate than they actually are. I'm also concerned that the author's viral detection pipeline may be underpowered, given that only 4,709 genomes were identified from 4,198 metagenomes. Unless the datasets are not sequenced deeply, I would have expected a much greater number of viruses. Using a subset of the samples, how many viral genomes are identified using an alternative, well-benchmarked pipeline, such as VIBRANT? CheckV was run to evaluate completeness, but did the authors also use it to remove contamination from integrated prophages? Lastly, the authors applied a 10 Kb length threshold to identify phage genomes, but this cutoff will miss most if not all of the microviruses and CRESS DNA viruses, which may represent a considerable % of reads from the VLP metagenomes – this should be noted in the text.

Regarding the analysis of novel phage groups – it would be helpful to cite additional studies here. Shah et al. doi:10.1101/2021.07.02.450849v2 were the first to conduct a large-scale, systematic taxonomic analysis of all the major groups in the gut virome and Figure 2A in the current manuscript and Figure 4B in Shah et al. are related. Benler et al. should also be cited here, as these authors described three new major clades (Quimbyviridae, Gratiaviridae, Flandersviridae) as well as Cornuault et al. who carefully described a novel group of *F. prausnitzii* phages that carry DGRs.

I appreciated the read mapping analysis of VLP metagenomes, especially that the authors removed both human and bacterial sequences prior to vOTU mapping. One question I have is whether the unmapped VLP reads represent viruses that were (1) present in the bulk metagenome, but were not detected by the viral identification pipeline, (2) present in the bulk metagenome, but were not abundant enough to assembly into 10 Kb contigs, (3) were not present in the bulk metagenome (e.g. due to size filtering). The authors should be able to address this point by mapping VLP reads to a non-redundant set of contigs from the bulk metagenomes, before and after filtering contigs by the 10Kb length cutoff.

Regarding the mapping analysis presented in Figure 2A, can the authors report what % of bulk metagenomic reads, and what % of VLP reads mapped to crass-like phages? In a recent paper, Yutin et al. (doi: 10.1038/s41467-021-21350-w) estimated that crass-like viruses recruit ~86% of viral reads from bulk metagenomes. It would be interesting to compare this number with what you observed.

I thought it was interesting that the abundance of generalist phages was less well correlated with single host taxa but I was a little surprised that most virus-host correlations were positive (rather than negative). I would expect positive correlations for temperate phages, but negative correlations for lytic phages. Can the authors report correlation patterns for lytic and temperate phage? Related to this, it was unclear how the authors are classifying their phages as lytic vs temperate. Several recent tools are capable of this, including BACPHILIP, phageAI, and DeePhage.

Figure 3e presents a very interesting result in which the abundance of defense systems is positively correlated with phage diversity. I can't help but wonder if there are any other explanations for this pattern. Is the abundance of defense systems directly correlated with the abundance of other bacterial genes, such as the ribosome? If this were the case, then defense system abundance would simply be reflecting the overall % of bacterial reads.

Minor comments:

Regarding the paper title, it was not immediately clear whether "population-level" is referring to viral populations or human populations.

Line 134: Note Flandersviridae and Gubaphage are the same taxonomic group

Reviewer #2 (Remarks to the Author):

The manuscript describes an impressive catalogue of bacteriophage genomic sequences extracted from gut microbiome whole-community metagenomic data. The bioinformatics methodology used in this study is adequate to the task and is well described.

The degree of novelty of this viral catalogue, relative to a number of several recently published similar studies is hard to assess, as only pairwise comparisons of the catalogue with existing databases were reported in the study.

However, the authors also analysed associations between virome composition on one hand, and bacteriome composition and host factors on the other, which adds value to this study. Conclusions are well supported by the data.

Based on my review of this manuscript, I recommend minor revision.

I have several specific comments which should be relatively easy to address.

Line 25: as references for phage-mediated HGT I would suggest including:

<https://doi.org/10.1186/s40168-020-00935-5>

Line 28: crAssphage and related crAss-like phages constitute a single large clade of viruses. See

<https://doi.org/10.1038/s41467-021-21350-w>

Line 56: It would be useful to indicate age range (in addition to median age), and perhaps numbers of subjects in different age groups: children, young adults, middle-aged, elderly, etc.

Line 86: What was the number of novel vOTUs relative to all published databases taken together (RefSeq+IMG/VR+GVD+GPD+MGV)? Extended Data Fig. 3 only shows pairwise Venn diagrams.

Line 90: While CRISPR-base host prediction is a robust enough technique, it still leaves opportunity for mistakes due to spurious matches. I see high percentage of phages with history of infection of different bacterial genera as suspicious. Such phages are rare among the cultured phages.

Lin 153: Viral taxonomic names only become official when a proposal to create such names is ratified by International Committee for Taxonomy of Viruses (ICTV). The names proposed here, therefore should not be italicised and should be given with quotation marks.

Line 163: " β -diversity of the virome was significantly higher than that of the bacteriome" - at what taxonomic level? I'm not sure that these two beta-diversity metrics are directly comparable...

Line 167: Is Spearman correlation (if I'm expanding r_s correctly) the right method in this case? I would highly recommend to consult with the following study

(<https://doi.org/10.1038/ismej.2015.235>) which analyses performance of difference correlation techniques in context of various models of ecological interaction in the microbiome.

Line 169: Was there any attempt made to predict life styles in these phages? Could it be phages with tighter correlation to their bacterial hosts are temperate phages? Or even prophages, directly associated with host genomes?

Line 297: It would make sense to also deposit these genomes (with annotations) to NCBI GenBank.

Thank you for inviting us to submit a revised draft of our manuscript entitled "Extensive gut virome variation and its associations with host and environmental factors in a population-level cohort" to Nature Communications. We appreciate the time and effort each of the reviewers have dedicated to providing insightful comments on ways to strengthen our paper. We have incorporated additional analyses and results reflecting the invaluable suggestions and comments of the reviewers in the revised manuscript. We hope that the revised manuscript and our responses provided below satisfactorily address all of the issues and concerns which the reviewers have noted.

Reviewer #1 (Remarks to the Author):

In their manuscript, "Population-level metagenomic exploration of gut virome variation", the authors conduct an in depth analysis of the gut virome using 4,198 whole metagenomes of deeply phenotyped Japanese individuals. The authors developed their own bioinformatics pipeline to discover 1,125 complete and 3,584 draft viral genomes, which the authors estimate cover 57% of virome abundance based on mapping reads from VLP metagenomes. Clustering of the data revealed 1,347 viral operational taxonomic units, and these are explored for novel taxa. The authors used extensive phenotyping data to identify host and environmental factors that correlate with the gut virome. I found the study to be well conducted, well written, with excellent quality figures.

Major comments:

I have several comments and questions regarding the phage pipeline. First, it's unclear to me why a novel pipeline was needed for the current study, given the large number of recently published tools, including Virsorter 2, VIBRANT, viralVerify, and Cenote Taker 2, which all make use of viral marker genes, and several of which can handle provirus sequences. None of these tools are evaluated in the benchmark (extended data figure 1), whereas, I found it strange to benchmark and report accuracy for CheckV which is not a viral detection pipeline – this makes it seem as though existing tools are less accurate than they actually are. I'm also concerned that the author's viral detection pipeline may be underpowered, given that only 4,709 genomes were identified from 4,198 metagenomes. Unless the datasets are not sequenced deeply, I would have expected a much greater number of viruses. Using a subset of the samples, how many viral genomes are identified using an alternative, well-benchmarked pipeline, such as VIBRANT?

Response: Thank you for pointing out this. We have newly added Virsorter2, VIBRANT, Seeker, and ViralVerify to the benchmarking and compared their performances with our custom pipeline. We found that the true positive ratio of these tools (55.9%–99.7%) was comparable with ours (81.5%) but false positives were still higher (2.6–54.2%) than with our pipeline (0.4%). In this study, we used bulk (whole) metagenomes, in which the majority of reads derive from non-phage sequences, such as bacterial chromosomes and plasmids (could be ~99% of sequences), rather than phage sequences (approximately 1%). Therefore, a phage catalog could be substantially contaminated by these non-phage sequences if there are many false positive predictions. To reduce such contamination as much as possible, we used the custom pipeline with the lowest false positive rate with relatively strict criteria.

To compare our results with those based on the other tool, we used VIBRANT on the 4,198 whole metagenomic dataset. We identified 99,032 genomes in total as raw output from VIBRANT (with the default option). The number of genomes decreased to 6,336 after we excluded low-quality genomes (as defined by VIBRANT or <70% completeness by checkV) and those with possible contaminations (>10% contamination by checkV). Although the number is still approximately 1.5 times higher than that from our custom pipeline (n = 4,709), we identified some genomes with possible contaminations in the VIBRANT-based catalog. In addition, to check the coverage of viral sequences by the VIBRANT-based catalog, we mapped the VLP reads to the 6,336 genomes and found that 58% of the reads were mapped, on average, which is only 1% higher than results based on our 4,709 genomes (57.1%). These results suggest that the coverage of viral sequences by the VIBRANT-based catalog is compatible with our catalog, but the former could be more contaminated by non-phage sequences. Therefore, we concluded that our phage catalog was still appropriate for the downstream analysis in the study. Also, it should be noted that Benler et al¹ also used strict criteria to detect phages in bulk metagenomes, and their number of detected phages was comparable to ours (3,738 phages in 5,742 bulk metagenomes).

We have updated the figures for the benchmarking according to the reviewer's comment (**Extended Data Fig. 1bc**) and added descriptions to the results and methods (line 66–69, line 405–407, and 457–458).

CheckV was run to evaluate completeness, but did the authors also use it to remove contamination from integrated prophages?

Response: We used checkV not only to evaluate completeness, but also to estimate contamination of draft genomes, and excluded drafts if they had >10% contamination (**Methods**).

Lastly, the authors applied a 10 Kb length threshold to identify phage genomes, but this cutoff will miss most if not all of the microviruses and CRESS DNA viruses, which may represent a considerable % of reads from the VLP metagenomes – this should be noted in the text.

Response: Thank you for raising this point. As the reviewer points out, the 10 Kb length threshold could remove small viruses in the human gut. However, we used bulk metagenomes in this study, and the main target is double-stranded DNA (dsDNA) viruses whose genome sizes are typically larger than 10 Kb in length². Although small viruses, such as single-stranded DNA viruses including microviruses and CRESS DNA viruses, are abundant in the VLP dataset with DNA amplification, these viruses are not sequenced with the standard methodology for bulk metagenomes³. Indeed, the 10 Kb length threshold would be necessary for the accurate identification of dsDNA phages⁴ and has also been used in several previous studies⁵⁻⁷. Therefore, we used the same threshold to construct a high-quality dsDNA phage genome catalog in this study. To emphasize these points, we have added new descriptions in the methods section in the manuscript (line 415–416).

Regarding the analysis of novel phage groups – it would be helpful to cite additional studies here. Shah et al. doi:10.1101/2021.07.02.450849v2 were the first to conduct a large-scale, systematic taxonomic analysis of all the major groups in the gut virome and Figure 2A in the current manuscript and Figure 4B in Shah et al. are related. Benler et al. should also be cited here, as these authors described three new major clades (Quimbyviridae, Gratiaviridae, Flandersviridae) as well as Cornuault et al. who carefully described a novel group of *F. prausnitzii* phages that carry DGRs.

Response: Thank you for pointing this out. We have cited these studies in the manuscript (line 125).

I appreciated the read mapping analysis of VLP metagenomes, especially that the authors removed both human and bacterial sequences prior to vOTU mapping. One question I have is whether the unmapped VLP reads represent viruses that were (1) present in the bulk metagenome, but were not detected by the viral identification pipeline, (2) present in the bulk

metagenome, but were not abundant enough to assemble into 10 Kb contigs, (3) were not present in the bulk metagenome (e.g. due to size filtering). The authors should be able to address this point by mapping VLP reads to a non-redundant set of contigs from the bulk metagenomes, before and after filtering contigs by the 10Kb length cutoff.

Response: To explore this, we re-mapped the unmapped VLP reads to the non-redundant set of contigs from the 4,198 bulk metagenome dataset. We found that $69.5 \pm 21.3\%$ (mean \pm s.d.) of the unmapped VLP reads on average were mapped to the contigs (Figure below). To characterize these contigs ($n = 168,169$), we performed a quality assessment by checkV and found that 6.8%, 7.6% and 85.6% of the contigs were high-quality (including complete), medium-quality, and low-quality (including non-determined) phages, respectively. The majority of the contigs was low-quality, suggesting that these contigs had low abundance or repetitive regions in the genomes prevented assembly⁸. On the other hand, some of the contigs were medium-/high-quality genomes, suggesting that they were abundant enough for the assembly, but our pipeline failed to detect them as phages. It will be necessary to construct a more accurate pipeline to detect such phages to cover more varied viral sequence spaces in the human gut.

Regarding the mapping analysis presented in Figure 2A, can the authors report what % of bulk metagenomic reads, and what % of VLP reads mapped to crass-like phages? In a recent paper, Yutin et al. (doi: 10.1038/s41467-021-21350-w) estimated that crass-like viruses recruit ~86% of viral reads from bulk metagenomes. It would be interesting to compare this number with what you observed.

Response: The number of reads mapped to crAss-like phages accounted for 5.2% and 17.1% of total bulk and VLP reads mapped to the catalog, respectively. This number was substantially lower than that reported in Yutin et al. We hypothesize that this could be due to the difference in the catalog used for the estimation because the number is the proportion of reads mapped to the catalog, and therefore substantially dependent on the catalog to be used. For example, if the catalog is more biased for crAss-like phages and other phages are not included, a higher proportion of reads are mapped to crAss-like phages. To explore this, we checked the methodologies used in Yutin et al. While we could not find details for the catalog to be used for the estimate (e.g. the number of genomes, phage names, and genome sequences), we found the following description in the Methods section:

“To estimate the abundances, the reads from each sample were aligned against a database that consisted of all complete bacterial, archaeal, and viral genomes in RefSeq, with the addition of the crAss-like sequences identified in the present work.”

If this refers to the methodology used to estimate the proportion of crAss-like phages, the catalog appears designed to be highly biased for crAss-like phages, since the majority of phage genomes in the human gut are not deposited in the RefSeq database. Thus, the substantial difference between our result and that of Yutin et al could be due to the difference in the catalogs of the two studies.

I thought it was interesting that the abundance of generalist phages was less well correlated with single host taxa but I was a little surprised that most virus-host correlations were positive (rather than negative). I would expect positive correlations for temperate phages, but negative correlations for lytic phages. Can the authors report correlation patterns for lytic and temperate phage? Related to this, it was unclear how the authors are classifying their phages as lytic vs temperate. Several recent tools are capable of this, including BACPHILIP, phageAI, and DeePhage.

Response: Thank you for your comment on this. In the original manuscript, we defined phages as temperate phages if they encoded integrase or were mapped to RefSeq reference bacterial genomes (>1,000bp and >95% identity). In response to your comment, we re-analyzed phage-host correlations by using BACPHILIP, which defines temperate phages based on presence/absence of more broad hallmark genes such as integrase, excisionase, and recombinase. The results showed that lytic phages have only slightly, but significantly higher, correlations with their hosts ($P = 0.025$) than temperate phages. Since phages kill their hosts,

a negative correlation between phages and their hosts could be observed in time-wise analyses. Actually, several previous studies observed negative correlations between them in a time-series dataset using a mouse model^{9,10}. At the same time, however, phages cannot live without their hosts, suggesting that their symbiotic relationship could result in a positive correlation at a more global scale (e.g. among different individuals and environments). Therefore, we have concluded that the presence of bacterial hosts is a major determinant for the distribution of phages in the human gut in a population-level cohort. We have added these discussions in the discussion part (line 277–283), and results in **Supplementary Table 6** and **Extended Data Fig. 9b**.

Figure 3e presents a very interesting result in which the abundance of defense systems is positively correlated with phage diversity. I can't help but wonder if there are any other explanations for this pattern. Is the abundance of defense systems directly correlated with the abundance of other bacterial genes, such as the ribosome? If this were the case, then defense system abundance would simply be reflecting the overall % of bacterial reads.

Response: To investigate how other gene functions of the bacteriome are associated with virome diversity, we newly performed a correlation analysis between virome diversity and all KEGG orthologies in the bacteriome. We found that integrases and some other genes for spore germination were also significantly correlated with the virome diversity (line 197–199, **Supplementary Table 9**). Since the bacteriome contains numerous and varied gene functions, and they are correlated to each other, it is quite challenging to determine which gene functions actually increase the virome diversity and which are just confounded based on our cross-sectional design. To emphasize this point, we have added this as a limitation of our analysis in the discussion part (line 290–293).

Minor comments:

Regarding the paper title, it was not immediately clear whether “population-level” is referring to viral populations or human populations.

Response: Thank you for raising this point. We have changed the title to “Extensive gut virome variation and its associations with host and environmental factors in a population-level cohort”. We used the phrase “population-level cohort” to clarify that the word “population” refers to humans.

Line 134: Note Flandersviridae and Gubaphage are the same taxonomic group

Response: Accordingly, we have revised the sentence (line 29, 142, 268).

Reviewer #2 (Remarks to the Author):

The manuscript describes an impressive catalog of bacteriophage genomic sequences extracted from gut microbiome whole-community metagenomic data. The bioinformatics methodology used in this study is adequate to the task and is well described. The degree of novelty of this viral catalog, relative to a number of several recently published similar studies is hard to assess, as only pairwise comparisons of the catalog with existing databases were reported in the study. However, the authors also analysed associations between virome composition on one hand, and bacteriome composition and host factors on the other, which adds value to this study. Conclusions are well supported by the data. Based on my review of this manuscript, I recommend minor revision.

I have several specific comments which should be relatively easy to address.

Line 25: as references for phage-mediated HGT would suggest including:
<https://doi.org/10.1186/s40168-020-00935-5>

Response: Accordingly, we have added the reference (line 25).

Line 28: crAssphage and related crAss-like phages constitute a single large clade of viruses. See <https://doi.org/10.1038/s41467-021-21350-w>

Response: Thank you for raising this. We have modified the description to clarify that crAssphage and crAss-like phages are in a single clade (line 28).

Line 56: It would be useful to indicate age range (in addition to median age), and perhaps numbers of subjects in different age groups: children, young adults, middle-aged, elderly, etc.

Response: Accordingly, we have added the standard deviation and age groups in the manuscript (line 55–57).

Line 86: What was the number of novel vOTUs relative to all published databases taken together (RefSeq+IMG/VR+GVD+GPD+MGV)? Extended Data Fig. 3 only shows pairwise Venn diagrams.

Response: Thank you for pointing out this. We have analyzed this and found that 667 vOTUs out of 1,347 vOTUs were still unique to our dataset. We have added the description of this result (line 91–92) and updated the **Extended Data Fig. 3**.

Line 90: While CRISPR-base host prediction is a robust enough technique, it still leaves opportunity for mistakes due to spurious matches. I see high percentage of phages with history of infection of different bacterial genera as suspicious. Such phages are rare among the cultured phages.

Response: As the reviewer points out, we cannot rule out the possibility of some false positives in our host predictions. However, based on the scope of our analytical methods and previous studies, our results appear to be reasonable. For example, in our host predictions, 63.3% of the phages were assigned to at least one host. This prediction rate is much higher than those of previous studies (28.7% for Camarillo-Guerrero 2021⁷, 29.0% for Benler 2021¹). This could be because we used a much larger bacterial genome database in the RefSeq than the previous studies, as well as assembled contigs (**Methods**). Therefore, the large-scale CRISPR-spacer dataset from the genomes/contigs enabled us to predict a wide range of phages' hosts despite the strict threshold for alignment (100% identity or only 1 mismatch allowed), which could result in the identification of more phages infecting multiple hosts beyond one genus. Indeed, several review articles have suggested that such generalist phages are more common in environments than was expected previously^{11,12}, and some studies have identified and succeeded in culturing such broad-host range phages^{13,14}. In addition, a study that used a single-cell technique to explore phage-bacteria interaction¹⁵ also detected phages that infect different genera in Firmicutes, which is consistent with our results.

Lin 153: Viral taxonomic names only become official when a proposal to create such names is ratified by International Committee for Taxonomy of Viruses (ICTV). The names proposed here, therefore should not be italicised and should be given with quotation marks.

Response: Thank you for pointing out this. According to the reviewer's comment, we have revised the viral taxonomic names throughout the manuscript (line 161–163).

Line 163: "β-diversity of the virome was significantly higher than that of the bacteriome" - at what taxonomic level? I'm not sure that these two beta-diversity metrics are directly comparable...

Response: Thank you for raising this point. To make it clearer, we have added a plot comparing β-diversity between the virome and bacteriome (**Extended Data Fig. 8**). We used a vOTU-level (corresponding the species level of phages) profile for the virome and a species-level profile for the bacteriome in this analysis. We have added this information in the figure legend.

Line 167: Is Spearman correlation (if I'm expanding rs correctly) the right method in this case? I would highly recommend to consult with the following study (<https://doi.org/10.1038/ismej.2015.235>) which analyses performance of difference correlation techniques in context of various models of ecological interaction in the microbiome.

Response: Thank you for pointing this out. We used Spearman correlation for this analysis because this method is robust for outliers and does not assume a normal distribution that is unlikely in the metagenomic data. Indeed, Spearman correlation coefficient is commonly used in studies to evaluate the associations between phages and their bacterial hosts^{16–18}. To check how methodologies affect the results, we reanalyzed the data with other indexes such as Pearson correlation, maximal information coefficient (MIC), and Bray-Curtis dissimilarity. Consistent with the results based on Spearman correlation, we found that both Pearson correlation and MIC showed positive correlations between phages and their hosts, in which specialist phages were more strongly correlated with the hosts than generalist phages. On the other hand, Bray-Curtis dissimilarity showed that generalist phages are more strongly correlated with their hosts than specialist phages. We have added this result to **Extended Data Fig. 9a** and descriptions to results (line 183–186).

Line 169: Was there any attempt made to predict life styles in these phages? Could it be phages with tighter correlation to their bacterial hosts are temperate phages? Or even prophages, directly associated with host genomes?

Response: By predicting phage lifestyles using BACPHLIP, we performed the same correlation analysis separately for lytic and temperate phages. The results showed that lytic phages correlated only slightly, but more strongly, with their hosts than temperate phages ($P = 0.025$). We have added this result to **Extended Data Fig. 9b**. We have added this result to the results (line 186–188)

Line 297: It would make sense to also deposit these genomes (with annotations) to NCBI GenBank.

Response: According to the reviewer's comment, we have deposited the viral genomes identified in this study ($n = 4,709$) to NCBI GenBank (accession number is PRJNA684915, available upon publication). The viral genomes are also available in Zenodo (<https://doi.org/10.5281/zenodo.5645361>).

References

1. Benler, S. *et al.* Thousands of previously unknown phages discovered in whole-community human gut metagenomes. *Microbiome* **9**, 78 (2021).
2. Hatfull, G. F. & Hendrix, R. W. Bacteriophages and their genomes. *Curr. Opin. Virol.* **1**, 298–303 (2011).
3. Roux, S. *et al.* Towards quantitative viromics for both double-stranded and single-stranded DNA viruses. *PeerJ* **4**, e2777 (2016).
4. Roux, S. *et al.* Minimum Information about an Uncultivated Virus Genome (MIUViG). *Nat. Biotechnol.* **37**, 29–37 (2019).
5. Nishimura, Y. *et al.* Environmental Viral Genomes Shed New Light on Virus-Host Interactions in the Ocean. *mSphere* **2**, (2017).
6. Emerson, J. B. *et al.* Host-linked soil viral ecology along a permafrost thaw gradient. *Nat Microbiol* **3**, 870–880 (2018).
7. Camarillo-Guerrero, L. F., Almeida, A., Rangel-Pineros, G., Finn, R. D. & Lawley, T. D. Massive expansion of human gut bacteriophage diversity. *Cell* **184**, 1098–1109.e9 (2021).
8. Kiguchi, Y., Nishijima, S., Kumar, N., Hattori, M. & Suda, W. Long-read metagenomics of multiple displacement amplified DNA of low-biomass human gut phageomes by SACRA pre-processing chimeric reads. *DNA Res.* **28**, dsab019 (2021).

9. Reyes, A., Wu, M., McNulty, N. P., Rohwer, F. L. & Gordon, J. I. Gnotobiotic mouse model of phage–bacterial host dynamics in the human gut. *Proceedings of the National Academy of Sciences* **110**, 20236–20241 (2013).
10. Hsu, B. B. *et al.* Dynamic Modulation of the Gut Microbiota and Metabolome by Bacteriophages in a Mouse Model. *Cell Host Microbe* **25**, 803–814.e5 (2019).
11. Ross, A., Ward, S. & Hyman, P. More Is Better: Selecting for Broad Host Range Bacteriophages. *Front. Microbiol.* **7**, 1352 (2016).
12. de Jonge, P. A., Nobrega, F. L., Brouns, S. J. J. & Dutilh, B. E. Molecular and Evolutionary Determinants of Bacteriophage Host Range. *Trends Microbiol.* **27**, 51–63 (2019).
13. Matilla, M. A. & Salmond, G. P. C. Bacteriophage ϕ MAM1, a viunalikevirus, is a broad-host-range, high-efficiency generalized transducer that infects environmental and clinical isolates of the enterobacterial genera *Serratia* and *Kluyvera*. *Appl. Environ. Microbiol.* **80**, 6446–6457 (2014).
14. Peters, D. L., Lynch, K. H., Stothard, P. & Dennis, J. J. The isolation and characterization of two *Stenotrophomonas maltophilia* bacteriophages capable of cross-taxonomic order infectivity. *BMC Genomics* **16**, 664 (2015).
15. Džunková, M. *et al.* Defining the human gut host–phage network through single-cell viral tagging. *Nature Microbiology* **4**, 2192–2203 (2019).
16. Dutilh, B. E. *et al.* A highly abundant bacteriophage discovered in the unknown sequences of human faecal metagenomes. *Nat. Commun.* **5**, 4498 (2014).
17. Norman, J. M. *et al.* Disease-specific alterations in the enteric virome in inflammatory bowel disease. *Cell* **160**, 447–460 (2015).
18. Waller, A. S. *et al.* Classification and quantification of bacteriophage taxa in human gut metagenomes. *ISME J.* **8**, 1391–1402 (2014).

REVIEWERS' COMMENTS

Reviewer #1 (Remarks to the Author):

I thank the authors for addressing my comments with insightful analysis, discussion, and figures. I have a few additional comments and suggestions below:

Regarding the benchmarking analysis, the authors write: "Comparison between our pipeline and other virus-detection tools showed that the true positive rate of our pipeline (81.5%) was comparable with other pipelines (40.2%–99.7%)". I feel a more accurate description is that the authors' pipeline was designed to be highly specific at the cost of lower sensitivity. This is clear from the comparison versus the three other marker-genes based pipelines (virsorter2, viralverify, and vibrant) as well as the comparison to Benler et al. and analysis of unmapped VLP reads. For example, Benler et al. identified 3,738 complete phage genomes in 5,742 bulk metagenomes (0.65 genomes/sample) versus the current study which found 1,125 complete genomes in 4,198 bulk metagenomes (0.27 genomes/sample).

I thank the authors for their comparison versus VIBRANT - is there a way to include this discussion in the supplementary text? When comparing to VIBRANT (# of genomes and % of VLP reads mapped), is it possible to use the same genome filtering criteria as applied to the current dataset for more of an apples-to-apples comparison? (the current dataset did not use VIBRANT completeness as a filtering parameter).

Thank you for clarifying your reasoning behind the 10Kb threshold to focus on dsDNA phages. It would be helpful to readers to clarify this in the abstract: "Here, we analyse dsDNA phages in a population-level cohort of ..."

I found the analysis of unmapped VLP reads to be interesting. Is it possible to include the figure in the supplement? And likewise with the comparison of crass-like abundance in Yutin et al.

It's interesting that that the authors found genes related to sporulation and integration were positively associated with phage diversity. This is likely reflecting the fact that Firmicutes often contain sporulation genes and have very diverse phages that tend to be temperate (Shah et al 2021).

Reviewer #2 (Remarks to the Author):

Many thanks for addressing all my comments! The manuscript has been considerably improved and I recommend it for publication in its current version.

We would like to thank the reviewers for the positive feedback and constructive suggestions that raised relevant and interesting suggestions to further strengthen our study. We hope that the revised manuscript and our responses provided below satisfactorily address all of the issues and concerns which the reviewers have noted.

Reviewer #1 (Remarks to the Author):

I thank the authors for addressing my comments with insightful analysis, discussion, and figures. I have a few additional comments and suggestions below:

Regarding the benchmarking analysis, the authors write: “Comparison between our pipeline and other virus-detection tools showed that the true positive rate of our pipeline (81.5%) was comparable with other pipelines (40.2%–99.7%)”. I feel a more accurate description is that the authors' pipeline was designed to be highly specific at the cost of lower sensitivity. This is clear from the comparison versus the three other marker-genes based pipelines (virsorter2, viralverify, and vibrant) as well as the comparison to Benler et al. and analysis of unmapped VLP reads. For example, Benler et al. identified 3,738 complete phage genomes in 5,742 bulk metagenomes (0.65 genomes/sample) versus the current study which found 1,125 complete genomes in 4,198 bulk metagenomes (0.27 genomes/sample).

Response: Thank you for pointing this out. We have revised the paragraph accordingly. We have used the phrase “slightly lower sensitivity” because our pipeline still showed a higher sensitivity than some of the tools (e.g. VirSorter and Seeker) (line 117).

I thank the authors for their comparison versus VIBRANT - is there a way to include this discussion in the supplementary text? When comparing to VIBRANT (# of genomes and % of VLP reads mapped), is it possible to use the same genome filtering criteria as applied to the current dataset for more of an apples-to-apples comparison? (the current dataset did not use VIBRANT completeness as a filtering parameter).

Response: We have newly prepared Supplementary Text and added the discussion there. Regarding the thresholds, we used the thresholds to the VIBRANT-based phages (>70% completeness and <10% contamination by checkV) because we used similar thresholds in our custom pipeline by comparing to complete viral sequences in IMG/VR database (i.e., linear genomes need to be >70% in length as compared with the complete viral genomes, line 509). Although the methodologies to define the completeness of a phage genome was not exactly the

same between our pipeline and checkV, we used the thresholds to compare the results with the same condition as much as possible.

Thank you for clarifying your reasoning behind the 10Kb threshold to focus on dsDNA phages. It would be helpful to readers to clarify this in the abstract: “Here, we analyse dsDNA phages in a population-level cohort of ...”

Response: We have added the word “dsDNA” in the abstract accordingly (line 53).

I found the analysis of unmapped VLP reads to be interesting. Is it possible to include the figure in the supplement? And likewise with the comparison of crass-like abundance in Yutin et al.

Response: We have added paragraphs in the Supplementary Text and discussed the results there.

It's interesting that the authors found genes related to sporulation and integration were positively associated with phage diversity. This is likely reflecting the fact that Firmicutes often contain sporulation genes and have very diverse phages that tend to be temperate (Shah et al 2021).

Response: Thank you for raising this point. We have added this discussion in the manuscript (line 249).

Reviewer #2 (Remarks to the Author):

Many thanks for addressing all my comments! The manuscript has been considerably improved and I recommend it for publication in its current version.